# Spike-HTR: Spiking Neural Transformer
# for Handwritten Text Recognition

Xiubo Liang [1]   Jinxing Han [1]   Yuke Li [2]   Haoqi Zhu [2]   Yu Zhao [1]   Hongzhi Wang [1 2]

## Abstract

Handwritten Text Recognition (HTR) is computationally imbalanced in two ways: most image pixels are background, and many width-axis sequence positions are blank-dominated. This creates a mismatch for Spiking Neural Networks (SNNs): handwriting is observed as a static image, whereas spiking computation unfolds over timesteps. We propose Spike-HTR, a hybrid spiking recognizer that controls both the number of spiking steps and the number of width positions processed by the deep sequence mixer. To make a static image suitable for short-horizon spiking inference, InkCoder converts it into a coarse-to-fine input stream, where early steps cover broad stroke regions and later steps emphasize sharper stroke details. To reduce sequence computation, a CTC-guided length reducer keeps likely character or uncertain positions and compresses long blank-dominated stretches before deep mixing. With $T{=}2$, Spike-HTR trains only on target data, decodes without language models or lexicons, and reaches validation/test CERs of 3.5/5.4, 2.3/2.5, and 4.2/3.9 on IAM, LAM, and READ2016. Codes are available at https://github.com/QomolangmaH/SpikeHTR.

## 1. Introduction

Handwritten Text Recognition (HTR) spends much of its computation on positions that contain little or no character evidence. At the image level, informative ink occupies only a small fraction of the canvas, while most pixels are background. Nevertheless, modern recognizers still process the full image with dense convolutions or attention-style mixing, spending computation and memory traffic on regions with little recognition evidence (Shi et al., 2016; Vaswani et al., 2017; Dosovitskiy, 2020; Liu et al., 2021; Bautista & Atienza, 2022; Li et al., 2023). A second redundancy appears after two-dimensional image features are collapsed into a one-dimensional sequence along the width axis. Many positions correspond to gaps, margins, or weak evidence, yet they are still processed by the deep sequence mixer. This sequence level redundancy is especially visible in Connectionist Temporal Classification (CTC), the alignment loss commonly used for segmentation-free text recognition. In CTC, many width-axis positions are assigned to a blank label rather than to output characters; treating all positions with the same deep-mixing budget can over-allocate computation to blank-dominated spans.

Spiking Neural Networks (SNNs) offer an event-based computation model that can, in principle, exploit sparse activity (Maass, 1997; Merolla et al., 2014; Furber et al., 2014; Davies et al., 2018; Tavanaei et al., 2019; Neftci et al., 2019; Orchard et al., 2021; Yao et al., 2023). However, offline HTR is static: spiking dynamics unfold over time, but handwriting is observed as a single image. A common workaround expands the input into $T$ steps by repeating the image or sampling rate/Poisson spikes from it (Auge et al., 2021; Rueckauer & Liu, 2021). For handwriting, repetition is stable but redundant, while stochastic sampling can perturb thin strokes and small gaps that are difficult to recover with only a few timesteps.

Spike-HTR addresses these issues by controlling two explicit budgets. The first is the temporal budget $T$, the number of spiking timesteps used to process one static line image. The second is the sequence budget $\ell_b$, the number of width-axis positions sent to the deep one-dimensional mixer after length reduction. For the temporal budget, we introduce InkCoder, a deterministic input module that converts one static image into a short coarse-to-fine sequence for spiking inference. Early steps preserve broad stroke regions, and later steps emphasize sharper stroke details, so timesteps are used for refinement rather than replaying the same image. For the sequence budget, we first use a lightweight CTC prediction to identify likely character positions and ambiguous positions along the width axis. The reducer keeps these positions and compresses long stretches that are likely to

---

[1]School of Software Technology, Zhejiang University, Ningbo, China [2]NetEase Yidun AI Lab, Hangzhou, China. Correspondence to: Hongzhi Wang <hongzhiwang@zju.edu.cn>.

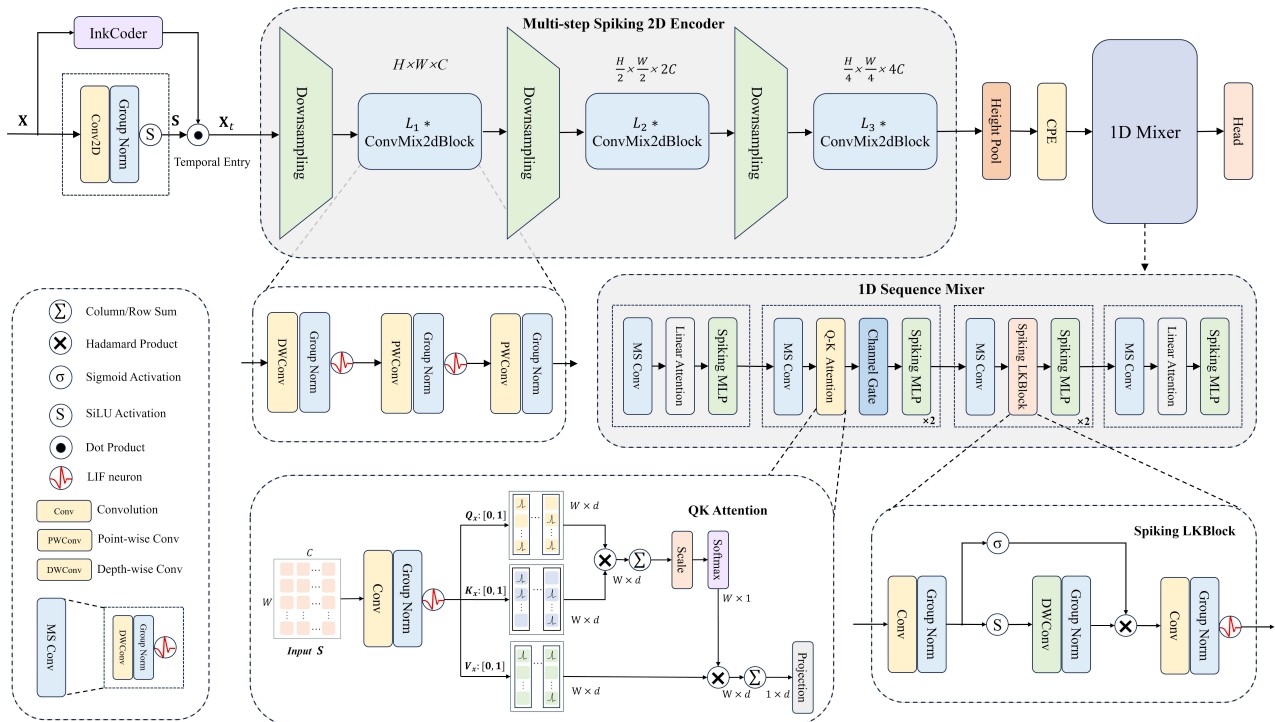

*Figure 1.* Overview of Spike-HTR architecture. A lightweight stem first extracts a shared feature map from the input line image. InkCoder derives a short sequence of spatial gates from the static image: early steps preserve broad stroke regions, and later steps emphasize sharper stroke details. A spiking 2D encoder extracts features and pools them over height to form a 1D sequence along the image width. Before the deep 1D mixer, a lightweight predictor keeps likely character or uncertain positions and compresses long blank regions. The shortened sequence is then mixed, fused across timesteps, and decoded with a CTC classifier.

be blank before the deep 1D mixer. This order-preserving reduction avoids treating all blanks as disposable, since short CTC blank intervals can still be needed to separate neighboring or repeated characters.

These two components have different scopes. The CTC length reducer is not tied to spiking: it only requires a CTC head that scores width positions, and can therefore also be attached to ANN-CTC recognizers. InkCoder is the spiking-specific component: it gives static handwriting a meaningful short temporal stream, making small-$T$ spiking inference a coarse-to-fine refinement process rather than repeated or randomly sampled evidence. On current dense accelerators, the most direct compute lever is the reduced width-axis length processed by the deep mixer. Beyond this, the spiking pathway targets neuromorphic deployment, where sparse event-driven execution can reduce activity and data movement when emitted spikes are few (Davies et al., 2018; Orchard et al., 2021). We therefore report spike-activity profiles to quantify the remaining event volume and assess how well the model preserves this hardware-facing sparsity.

Across IAM, LAM, and READ2016 (Marti & Bunke, 2002; Sanchez et al., 2016; Cascianelli et al., 2022), Spike-HTR reaches validation/test character error rates (CERs) of

3.5/5.4, 2.3/2.5, and 4.2/3.9 with $T$=2. Additional timesteps give diminishing CER gains, while the length reducer shortens the sequence before deep mixing with little accuracy loss. We report temporal-budget sweeps, reduction ablations, matched ANN controls, and spike-activity diagnostics to characterize accuracy, compute, and activity trade-offs.

Our contributions are:

- We formulate static-image spiking HTR as a two-budget problem: $T$ controls temporal refinement, and $\ell_b$ controls the width-axis sequence length processed by the deep mixer.

- We propose InkCoder, an input module that converts a static handwriting image into a short coarse-to-fine stream for low-timestep spiking inference.

- We introduce a CTC-guided length reducer that keeps likely character or uncertain positions and compresses long blank-dominated stretches before deep one-dimensional mixing.

- We build Spike-HTR, a spiking transformer recognizer trained without external pretraining or synthetic data, and analyze accuracy, sequence length, compute proxies, and spike activity across datasets.

## 2. Related Work

HTR has evolved from CNN–RNN pipelines to Transformer-based recognizers, often benefiting from large-scale pretraining or synthetic data (Shi et al., 2016; Bautista & Atienza, 2022; Li et al., 2023). These systems set strong accuracy baselines, but they typically treat compute as uniform over the canvas and over the width-axis sequence, despite handwriting being strongly blank-dominated. Our focus is complementary: we study mechanisms that explicitly budget computation for sparse stroke evidence (in space) and for blank-heavy CTC frames (along width), under a controlled from-scratch setting.

Advances in surrogate-gradient training and neuron dynamics have enabled deep SNN optimization through time (Neftci et al., 2019; Wu et al., 2019; Zheng et al., 2021; Fang et al., 2021; Meng et al., 2023). Spiking Transformers show that attention-style mixing can be made spike-compatible on dense vision benchmarks (Zhou et al., 2022; Yao et al., 2023; Shi et al., 2024; Zhou et al., 2024). For offline HTR, however, a key bottleneck is upstream: the input is static, so the model must obtain a short, faithful temporal stream to make small-$T$ spiking inference meaningful.

Static-to-spike conversions include frame repetition, stochastic rate/Poisson coding, and strict temporal codes such as time-to-first-spike (Auge et al., 2021; Rueckauer & Liu, 2021). For handwriting, repetition is stable but information-redundant; stochastic coding introduces variance that can perturb thin strokes; strict temporal codes can be brittle under low SNR. Recent work therefore begins to treat temporal entry as task-coupled rather than fixed. InkCoder follows this direction with deterministic, evidence-guided progressive gating designed specifically for short-horizon handwriting.

Token pruning and merging reduce redundant Transformer computation via dynamic sparsification and token merging (Rao et al., 2021; Liang et al., 2022; Bolya et al., 2023). In offline HTR, length reduction must respect CTC alignment: short blank separators and local timing cues can matter even when locally uninformative. Spike-HTR uses a blank-posterior preview to drive an order-preserving keep-and-merge reduction that compresses long blank-dominated runs into bounded spans while retaining a minimum number of positions for feasibility. Unlike generic token merging, our reducer is coupled to the CTC blank posterior and uses a stop-gradient controller so the reduction rule does not backpropagate through the preview decision.

## 3. Method

Spike-HTR is a spiking Transformer for offline handwritten text recognition with two explicit budgets: the spiking horizon $T$ and the post-reduction token length $\ell_b$ (the number of

CTC frames kept along the width axis). Increasing $T$ admits evidence through additional coarse-to-fine refinement steps and affects spike activity, whereas $\ell_b$ directly controls the sequence length seen by the deep 1D mixer. InkCoder injects a shared static stem feature into the spiking backbone via scheduled spatial gates, and a stop-gradient blank-posterior preview (reusing the shared CTC head, stabilized by cross-depth alignment) filters blank-dominated positions to set $\ell_b$ per sample. We study the temporal budget via controlled $T$-sweeps, the token budget via reduction ablations and the measured reduction ratio, and the activity budget via firing statistics (Section 4 and Appendix). Figure 1 summarizes the full computation graph. Figure 2 expands the InkCoder block and shows how a static text-line image is converted into timestep-indexed spatial gates. Token-length reduction is defined in Section 3.4 and is applied before the deepest 1D stack.

### 3.1. Preliminaries

Let $\mathbf{X} \in [0,1]^{B \times C \times H \times W}$ denote a normalized text-line image batch and let $Y^{(b)} = (y_1^{(b)}, \ldots, y_{U_b}^{(b)})$ be the transcription for sample $b$ over a vocabulary $\mathcal{V}$ augmented with the CTC blank symbol $\phi$ (id 0). After sequence modeling and temporal fusion, the model produces width-axis framewise logits $\mathbf{O}^{(b)} \in \mathbb{R}^{\ell_b \times |\mathcal{V}|}$ and probabilities $\mathbf{P}^{(b)} = \mathrm{Softmax}(\mathbf{O}^{(b)})$. For batched computation, variable lengths are padded to $L = \max_b \ell_b$ and stored as $\mathbf{O} \in \mathbb{R}^{B \times L \times |\mathcal{V}|}$ together with valid lengths $\{\ell_b\}$.

We use multi-step leaky integrate-and-fire (LIF) units with surrogate gradients. For neuron $i$ at step $t \in \{1, \ldots, T\}$, membrane dynamics follow

$$
\begin{aligned}
u_{t,i} &= \tau u_{t-1,i}\left(1 - s_{t-1,i}\right) + x_{t,i}, \\
s_{t,i} &= \Theta(u_{t,i} - V_{\mathrm{th}}),
\end{aligned}
\tag{1}
$$

where $\tau$ is the decay factor, $x_{t,i}$ is the synaptic input current, and $\Theta(\cdot)$ is the Heaviside step function. Multi-step blocks maintain recurrent membrane states over timesteps for each layer, and states are reset between samples in implementation. We index both spiking steps and InkCoder gates by $t = 1, \ldots, T$, and gate $g_t$ modulates the injected drive at step $t$.

We optimize Spike-HTR with the Connectionist Temporal Classification loss, which marginalizes over monotonic alignments $\pi \in (\mathcal{V} \cup \{\phi\})^{\ell_b}$:

$$
\mathcal{L}_{\mathrm{ctc}} = -\log \sum_{\pi \in \mathcal{B}^{-1}(Y^{(b)})} \prod_{k=1}^{\ell_b} \mathbf{P}_{k,\pi_k}^{(b)},
\tag{2}
$$

where $\mathcal{B}$ collapses repeats and removes blanks. In Spike-HTR, the nominal pre-reduction length is $L_0 \approx W/4$ due to two strided downsamplings, and the effective length

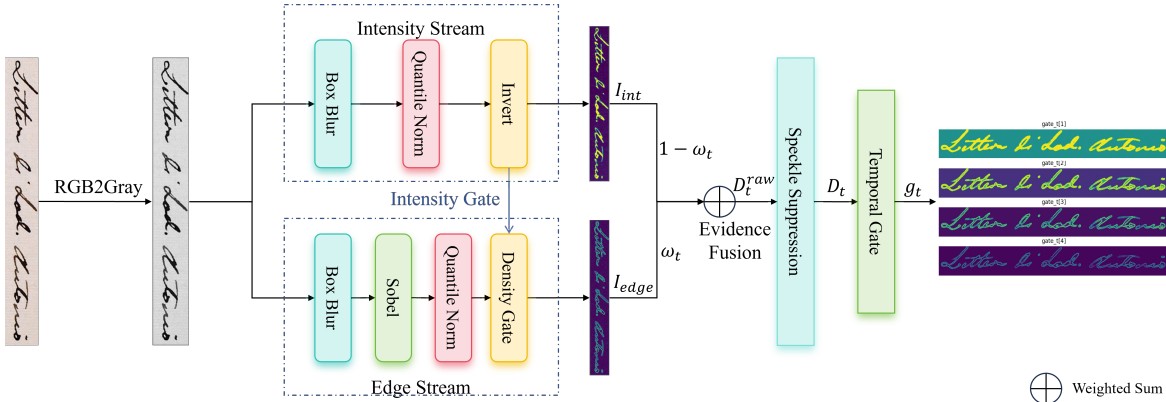

*Figure 2.* Mechanism of InkCoder. A static text-line image is first converted to grayscale and decomposed into two deterministic evidence streams. The intensity stream estimates stable stroke mass through smoothing, quantile normalization, and inversion. The edge stream extracts boundary-sensitive evidence and suppresses texture-induced responses using ink-supported and spatially coherent filtering. At timestep $t$, the two streams are fused with a scheduled weight $w_t$ to form raw evidence $D_t^{\text{raw}}$. Speckle suppression removes isolated artifacts from the fused evidence, and the temporal gate maps the stabilized evidence $D_t$ to the spatial gate $g_t$. Across timesteps, InkCoder turns one static image into a coarse-to-fine gated drive for short-horizon spiking inference.

$\ell_b \leq L_0$ is determined dynamically per sample by sparsity control.

## 3.2. InkCoder

Offline handwriting is observed as a single static image, whereas spiking computation unfolds over discrete timesteps. InkCoder resolves this static-to-temporal bottleneck by mapping one line image to a short sequence of spatial gates $\{g_t\}_{t=1}^T$. The gates inject stroke evidence in a deterministic coarse-to-fine order: early steps admit broad, stable stroke support, while later steps tighten selectivity and emphasize fine boundaries. This targets small-$T$ inference, where later steps refine ambiguous regions rather than replay the same image or amplify background texture.

As shown in Figure 2, InkCoder separates the temporal entry into three stages. First, it constructs complementary stroke evidence. The intensity stream provides a stable estimate of ink mass and is robust to mild illumination and thickness variation. The edge stream recovers thin strokes, small gaps, and junction topology, but filters its responses using local ink support and spatial coherence so that paper texture and isolated background edges are not amplified. Second, the two streams are fused over time: early steps emphasize intensity-supported stroke regions, while later steps increase the contribution of edge evidence. Third, density-based speckle suppression and a temporal gate convert the fused evidence into the spatial gate $g_t$ that modulates the drive injected into the spiking encoder. Thus, the timestep budget is used for structured refinement instead of repeated static input.

InkCoder is deterministic in its evidence construction and lightweight; only two global gate-sharpness scalars are learned. It encodes structural priors of handwriting: strokes are spatially coherent, useful edges should be supported by nearby inkness, and isolated responses are likely noise. Encoding these priors directly yields a stable temporal entry and avoids the instability of learning dynamic gating under short temporal simulation.

Let $X_{\text{gray}} \in [0,1]^{B \times 1 \times H \times W}$ be the grayscale image derived from $\mathbf{X}$. InkCoder constructs two bounded evidence proxies in $[0,1]^{B \times 1 \times H \times W}$: an intensity proxy $I_{\text{int}}$ and an edge proxy $I_{\text{edge}}$. The intensity proxy stabilizes stroke mass under illumination and thickness variation. The edge proxy recovers thin strokes, small gaps, and junction topology. To avoid texture-induced edges, $I_{\text{edge}}$ is filtered by two suppressors with distinct roles: a cross-modal consistency gate that trusts edges primarily where intensity also indicates ink, followed by a spatial-coherence gate that suppresses isolated speckles and background texture. All proxy definitions, including multiscale edge construction, are provided in Section A.1.

InkCoder allocates proxy emphasis over time via a step-dependent convex fusion:

$$D_t^{\text{raw}} = \text{clip}\big((1-w_t)I_{\text{int}} + w_t I_{\text{edge}}, 0, 1\big), \qquad t = 1, \ldots, T, \tag{3}$$

where $w_t$ increases with $t$ so early steps favor stable intensity evidence, while later steps progressively incorporate edge detail. Each $D_t^{\text{raw}}$ is further stabilized by a density-based suppressor that removes speckle-like artifacts while preserving coherent strokes (Section A.1). We denote the stabilized evidence by $D_t \in [0,1]^{B \times 1 \times H \times W}$.

InkCoder converts stabilized evidence into gates via scheduled soft-thresholding:

$$g_t = \sigma(a_t(D_t - \theta_t)), \qquad t = 1, \ldots, T. \tag{4}$$

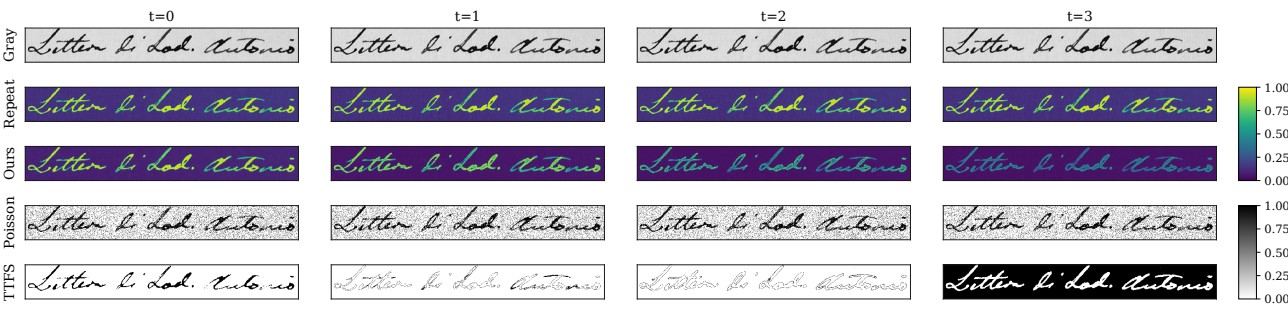

*Figure 3.* Temporal input streams under different static-to-temporal entry schemes (LAM sample #000883). Columns correspond to timesteps $t=1,\dots,T$. Gray shows the input reference; rows illustrate repeat injection, InkCoder, Poisson/rate coding, and latency coding. InkCoder produces the coarse-to-fine behavior induced by Figure 2: early steps preserve broad stroke support, while later steps tighten selectivity and sharpen stroke-aligned evidence. Repeat injection is temporally redundant, whereas Poisson/rate and latency drives can introduce speckle, frame-to-frame jitter, or discretization artifacts under scanned backgrounds.

Here $a_t$ controls how sharply the gate transitions around $\theta_t$. The gate threshold follows a monotone raw threshold schedule:

$$\lambda_t = \frac{t-1}{\max(T-1,1)}, \qquad \theta_t = \theta_{\min} + (\theta_{\max} - \theta_{\min})\lambda_t^{\gamma_\theta}. \tag{5}$$

This schedule admits broad stroke evidence at early steps and progressively tightens the gate at later steps.

### 3.3. Spiking Encoder

As summarized in Figure 1, the encoder factorizes static spatial extraction from time-stepped spiking processing. Given $\mathbf{X}$, a lightweight non-spiking stem produces a shared spatial representation

$$\mathbf{S} = \mathrm{SiLU}\big(\mathrm{Norm}(\mathrm{Conv}_{3\times3}(\mathbf{X}))\big) \in \mathbb{R}^{B \times c_1 \times H \times W}. \tag{6}$$

Temporal variation is introduced solely via InkCoder gates.

At each spiking step $t \in \{1,\dots,T\}$, we inject $\mathbf{S}$ via a gated drive with a non-zero baseline:

$$\mathbf{X}_t = \mathbf{S} \odot \big(\beta + (1-\beta)\,g_t\big), \qquad t = 1,\dots,T, \tag{7}$$

where $\beta \in (0,1)$ prevents a fully silent drive under aggressive gating. For a stable sparsity trade-off, we parameterize $\beta$ as a learnable stage-wise scalar constrained to $(0,1)$. While the underlying convolutions remain dense on standard accelerators, suppressing the injected drive reduces downstream spiking activity, which is the relevant budget on event-driven substrates and is quantified by firing and sparsity profiling.

Figure 3 shows the resulting time-indexed drives $\mathbf{X}_t$ after applying the gates from Figure 2. Repeat injection keeps $\mathbf{X}_t$ constant, so multi-step spiking mainly integrates redundant evidence. Poisson/rate coding introduces sampling noise that appears as speckle and frame-to-frame jitter in low-ink regions. Latency coding assigns evidence to discrete firing

times and can make temporal allocation sensitive to scanned backgrounds and low-SNR strokes. In contrast, InkCoder deterministically gates the shared stem and yields a stable coarse-to-fine stream. Additional proxy and gate diagnostics are provided in Section A.2.

The 2D spiking backbone follows a three-stage hierarchy with two stride-2 transition blocks, producing feature maps $F_t^{(2)}$ and $F_t^{(3)}$ at $1/2$ and $1/4$ resolution. To counteract low-pass spiking integration and preserve fine stroke topology, each spiking convolution uses a membrane shortcut:

$$\begin{aligned}
\mathbf{H}_t &= \mathrm{Norm}(\mathrm{Conv}(\mathbf{X}_t)), \\
\mathbf{Y}_t &= \mathrm{LIF}(\mathbf{H}_t) + \alpha\,\mathbf{H}_t,
\end{aligned} \tag{8}$$

where $\alpha$ is learnable and initialized near 0. We optionally regularize $\alpha$ to remain small, keeping the pathway close to spike-native while retaining analog pre-activations only when beneficial.

To retain higher-resolution morphology, Stage-2 details are injected into Stage-3 via dual-resolution fusion:

$$\begin{aligned}
\hat{F}_t^{(2)} &= \mathrm{DownMix}(\mathrm{Proj}(F_t^{(2)})), \\
F_t^{(3)} &\leftarrow F_t^{(3)} + \sigma(\mathrm{Gate}(F_t^{(3)})) \odot \hat{F}_t^{(2)},
\end{aligned} \tag{9}$$

where $\mathrm{Proj}$ is a $1\times1$ projection and the content-conditioned gate modulates how much higher-resolution detail is injected into Stage-3. The downsampling operator is a learnable mixture of average- and max-pooling: $\mathrm{DownMix}(z) = \mathrm{Avg}(z) + \rho(\mathrm{Max}(z) - \mathrm{Avg}(z))$ with $\rho \in (0,1)$, biasing the transition toward edge-preserving downsampling when needed.

We collapse the final 2D map into a 1D token sequence by pooling over height. Let $H' = H/4$ and $\mathbf{f}_{t,w} \in \mathbb{R}^{d \times H'}$ be the feature slice at width index $w$. We compute a learnable attention vector $\omega_{t,w} = \mathrm{Softmax}(\mathrm{Score}(\mathbf{f}_{t,w}))$ and a

channel-wise gate $\mathbf{r}_{t,w}$:

$$
\begin{aligned}
\mathbf{z}_{t,w}^{\text{attn}} &= \sum_{h=1}^{H'} \omega_{t,w,h}\, \mathbf{f}_{t,w,h}, \\
\mathbf{z}_{t,w}^{\max} &= \max_h \mathbf{f}_{t,w,h}, \\
\mathbf{z}_{t,w} &= \mathbf{z}_{t,w}^{\text{attn}} + \mathbf{r}_{t,w} \odot (\mathbf{z}_{t,w}^{\max} - \mathbf{z}_{t,w}^{\text{attn}}).
\end{aligned}
\tag{10}
$$

After pooling, the nominal token length is $L_0 \approx W/4$ due to the two strided transitions.

### 3.4. Sparsity Control

After height pooling, the encoder yields a time-indexed token stream $Z \in \mathbb{R}^{T \times B \times d \times L_0}$ with nominal length $L_0 \approx W/4$ (width-axis positions). Since handwriting CTC is typically blank-dominated, we reduce the effective length from $L_0$ to $\ell_b$ before deep 1D mixing so that the token-mixing budget tracks information density rather than canvas width.

**Position-aware preview.** We apply the same positional encoding used by the mixer:

$$
Z \leftarrow Z + P_{\text{abs}} + \text{CPE}(Z).
\tag{11}
$$

**Stop-gradient blank-posterior preview.** Reduction is driven by a stop-gradient preview computed with the shared classifier head. Let $\text{sg}(\cdot)$ be identity in the forward pass that blocks gradients. We implement $\text{Fuse}_{\text{pre}}(\cdot)$ as a learned convex fusion across timesteps:

$$
\begin{aligned}
\text{Fuse}_{\text{pre}}(Z) &= \sum_{t=1}^{T} \alpha_t^{\text{pre}}\, Z_t, \\
\boldsymbol{\alpha}^{\text{pre}} &= \text{Softmax}(\mathbf{a}^{\text{pre}}) \in \Delta^{T-1},
\end{aligned}
\tag{12}
$$

We fuse the $T$ spiking steps into a single preview sequence:

$$
\bar{Z} = \text{sg}(\text{Fuse}_{\text{pre}}(Z)) \in \mathbb{R}^{B \times d \times L_0}.
\tag{13}
$$

For sample $b$, let $\bar{\mathbf{z}}^{(b)}(w) = \bar{Z}_{:,w}^{(b)} \in \mathbb{R}^d$ for $w \in \{1, \ldots, L_0\}$. We estimate a per-position blank posterior via the shared head:

$$
\begin{aligned}
\mathbf{o}^{(b)}(w) &= \text{Cls}\Big(\text{Norm}\big(\bar{\mathbf{z}}^{(b)}(w)\big)\Big), \\
\pi^{(b)}(w) &= \text{Softmax}\Big(\mathbf{o}^{(b)}(w)\Big), \\
p_{\text{blank}}^{(b)}(w) &= \pi_\phi^{(b)}(w),
\end{aligned}
\tag{14}
$$

where $\phi$ denotes the CTC blank symbol. We also compute an entropy-based uncertainty score:

$$
u^{(b)}(w) = -\sum_{v \in \mathcal{V} \cup \{\phi\}} \pi_v^{(b)}(w)\, \log \pi_v^{(b)}(w).
\tag{15}
$$

Uncertain positions are forced to be retained (Eq. 16). To make shared-head preview reliable under shallow–deep distribution shift, we regularize shallow predictions with an auxiliary CTC loss and a masked cross-depth consistency term (Eq. 19).

**CTC-aware keep-and-merge reduction.** Given $p_{\text{blank}}$ and $u$, we perform an order-preserving keep-and-merge reduction that respects CTC topology: we keep likely non-blank singletons, preserve uncertain positions, and compress long blank runs into short spans. Define the keep set over original positions $w \in \{1, \ldots, L_0\}$:

$$
K^{(b)} = \left\{ w \,\middle|\, p_{\text{blank}}^{(b)}(w) < \tau \right\} \cup \left\{ w \,\middle|\, u^{(b)}(w) > \eta \right\}.
\tag{16}
$$

We enforce the label-free feasibility constraint $|K^{(b)}| \geq \lceil \gamma L_0 \rceil$ by adding positions with the smallest $p_{\text{blank}}^{(b)}(w)$ when needed. Traversing left-to-right, we form contiguous spans $\{S_m^{(b)}\}_{m=1}^{\ell_b}$: if $w \in K^{(b)}$ emit $S = \{w\}$; otherwise group consecutive blank-like positions into spans of size at most $k$.

**Span pooling (shared across timesteps).** We pool each span with a blank-aware weighted mean, differentiable w.r.t. token features:

$$
\begin{aligned}
a^{(b)}(w) &= 1 - p_{\text{blank}}^{(b)}(w), \\
\omega_m^{(b)}(w) &= \frac{a^{(b)}(w)}{\sum_{j \in S_m^{(b)}} a^{(b)}(j) + \delta}, \qquad w \in S_m^{(b)},
\end{aligned}
\tag{17}
$$

$$
\begin{aligned}
Z'_{t,b,:,m} &= \sum_{w \in S_m^{(b)}} \omega_m^{(b)}(w)\, Z_{t,b,:,w}, \\
t &= 1, \ldots, T, \quad m = 1, \ldots, \ell_b.
\end{aligned}
\tag{18}
$$

Here $\delta$ is a small constant for numerical stability. We pad to $L = \max_b \ell_b$ and carry valid lengths $\{\ell_b\}$ for masking and CTC input lengths. The reduction is a single left-to-right pass and costs $O(BL_0)$. Since the dominant 1D mixer cost scales with sequence length, we use the reduction ratio $r = \mathbb{E}[\ell_b/L_0]$ as a practical proxy for token-mixing compute under dynamic packing.

**Training losses and cross-depth alignment.** Decoding uses the main CTC head on the deep (post-mixer) stream. For reliable preview-driven reduction early in training, we also supervise the reduced pre-mixer stream with an auxiliary CTC loss using the same shared classifier head. We form a shallow fused sequence without stop-gradient, $\tilde{Z} = \text{Fuse}_{\text{pre}}(Z') \in \mathbb{R}^{B \times d \times L}$, and obtain auxiliary distributions on the padded reduced positions. Let $M_m^{(b)} = \mathbf{1}[m \leq \ell_b]$ be the valid-position mask. Let $P_{\text{main}}^{(b)}(m)$ and $P_{\text{aux}}^{(b)}(m)$ be the main and auxiliary distributions at reduced position $m$

*Table 1.* Main results on line level HTR (test CER, %) with greedy CTC. Spike-HTR rows follow our from-scratch protocol without external LM or lexicon. Prior ANN numbers may differ in data, pretraining, and decoding.

| Method | Type | Architecture | $T$ | Params(M) | IAM CER | LAM CER | READ2016 CER |
|---|---|---|---|---|---|---|---|
| CNN+BLSTM (Puigcerver, 2017) | ANN | CNN + BLSTM, CTC | – | 9.3 | 8.3 | 5.8 | 8.3 |
| Gated-CRNN (Bluche & Messina, 2017) | ANN | gated conv-RNN, CTC | – | 18.2 | 7.8 | 3.8 | 7.8 |
| PAR (Kang et al., 2022) | ANN | transformer (non-recurrent) | – | 54.7 | 4.7 | 10.2 | 4.7 |
| GFCN (Coquenet et al., 2020) | ANN | FCN-CTC | – | 1.4 | 8.0 | 5.2 | – |
| OrigamiNet (Yousef & Bishop, 2020) | ANN | FCN-CTC (+unfold) | – | 39.0 | 6.0 | 3.1 | – |
| VAN (Coquenet et al., 2022) | ANN | CNN + vertical attention | – | 2.7 | 5.0 | 4.1 | 4.1 |
| DAN (Coquenet et al., 2023) | ANN | doc attention net | – | 7.6 | – | – | 4.1 |
| SPAN (Coquenet et al., 2021) | ANN | FCN (SPAN enc.) + height pool, CTC | – | – | 4.8 | – | 4.6 |
| HTR-VT (Li et al., 2025) | ANN | CNN+ViT encoder, CTC | – | 53.5 | 4.7 | 2.8 | 3.9 |
| Spike-HTR (ours) | SNN | spiking transformer + CTC | 1 | 76.7 | 5.5 | 2.6 | 4.1 |
| Spike-HTR (ours) | SNN | spiking transformer + CTC | 2 | 76.7 | 5.4 | **2.5** | **3.9** |
| Spike-HTR (ours) | SNN | spiking transformer + CTC | 4 | 76.7 | 5.4 | **2.5** | 4.0 |

for sample $b$, and let $\mathrm{SKL}(p, q) = \frac{1}{2}(\mathrm{KL}(p\|q) + \mathrm{KL}(q\|p))$. We minimize

$$\mathcal{L} = \mathcal{L}_{\mathrm{ctc}}^{\mathrm{main}} + \lambda_{\mathrm{aux}}\mathcal{L}_{\mathrm{ctc}}^{\mathrm{aux}} + \lambda_{\mathrm{kl}}\mathcal{L}_{\mathrm{kl}},$$

$$\mathcal{L}_{\mathrm{kl}} = \frac{\sum_{b=1}^{B}\sum_{m=1}^{L} M_m^{(b)}\mathrm{SKL}\left(P_{\mathrm{main}}^{(b)}(m), P_{\mathrm{aux}}^{(b)}(m)\right)}{\sum_{b=1}^{B}\sum_{m=1}^{L} M_m^{(b)}}. \quad (19)$$

This alignment keeps shallow features CTC-decodable and calibrated under the shared head, enabling reliable preview-driven reduction.

### 3.5. 1D sequence mixer and temporal fusion

After reduction, we apply a stack of $N_{\mathrm{mix}}$ hybrid 1D mixer blocks to each timestep stream $Z_t' \in \mathbb{R}^{B \times d \times \ell_b}$ (padded to $L$ for batching). Each block composes three complementary token-mixing operators (details and ablations in Appendix C): (i) linear attention for stable global aggregation, (ii) a lightweight QK-style mixing for content gating, and (iii) large-kernel convolutional token mixing (Spiking LKBlock) for midrange continuity, followed by a spiking MLP. All operators use pre-norm and residual connections.

To produce CTC frame logits, we fuse the $T$ spiking steps into a single frame sequence with a learned convex temporal fusion:

$$Z^{\mathrm{fuse}} = \sum_{t=1}^{T} \alpha_t^{\mathrm{post}} Z_t^{\mathrm{deep}}, \quad (20)$$

$$\boldsymbol{\alpha}^{\mathrm{post}} = \mathrm{Softmax}(\mathbf{a}^{\mathrm{post}}) \in \Delta^{T-1},$$

where $Z_t^{\mathrm{deep}}$ denotes the post-mixer stream at step $t$. We then apply LayerNorm and a linear classifier to obtain logits $\mathbf{O}^{(b)} \in \mathbb{R}^{\ell_b \times |\mathcal{V}|}$ for CTC. For $T{=}1$, the fusion reduces to identity.

*Table 2.* Dataset statistics under the line level protocols used in our experiments.

| Dataset | Train | Val | Test | Lang. | Charset |
|---|---|---|---|---|---|
| IAM | 6,482 | 976 | 2,915 | English | 79 |
| LAM | 19,830 | 2,470 | 3,523 | Italian | 89 |
| READ2016 | 8,349 | 1,040 | 1,138 | German | 89 |

## 4. Experiments

We evaluate Spike-HTR on three line level HTR benchmarks with complementary recognition challenges: IAM for writer-disjoint English handwriting, LAM for long-term single-writer drift in Italian manuscripts, and READ2016 for degraded historical German handwriting. Table 2 summarizes the train/validation/test splits, languages, and character-set sizes used in our experiments. All Spike-HTR variants are trained from scratch on each target dataset without external pretraining, synthetic data, language models, or lexicons. Additional text normalization, preprocessing, base configuration, and training details are provided in Appendix B.

We report both character and word error rates (CER/WER) when comparing temporal budgets (Appendix Table 11), and use CER as the primary metric in the main table for space. To characterize activity-proportional computation, we summarize spike sparsity with module-wise firing rates and spike share (Figure 4); these statistics are hardware-agnostic proxies for energy on event-driven substrates and sparse kernels.

### 4.1. Main Results and Scaling with $T$

Table 1 reports test-set CER under this unified, from-scratch protocol. With $T{=}2$, Spike-HTR achieves 5.4 CER on IAM,

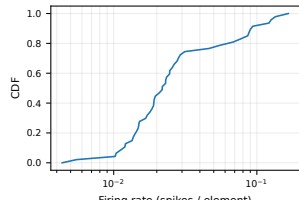 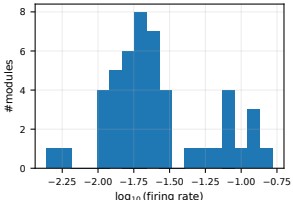 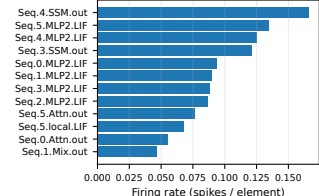 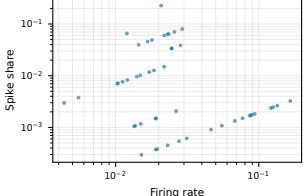

*(a)* CDF of module firing rates.  *(b)* Histogram of firing rates.  *(c)* Top modules by firing rate.  *(d)* Spike-share versus firing rate.

*Figure 4.* Event sparsity diagnostics for Spike-HTR on 512 IAM samples ($d$=768, $T$=2). Panels (a,b) show that module-wise firing rates concentrate at a few percent with a bounded tail, indicating globally sparse and well-controlled activity. Panel (c) localizes the highest firing intensities to a small subset of 1D mixer outputs. Crucially, panel (d) separates intensity from volume: despite higher firing rates in the 1D mixer, most spikes are produced by early 2D encoder layers because their activation maps are much larger.

*Table 3.* Isomorphic ANN control on IAM (validation CER, %).

| Variant | IAM |
|---|---|
| Spike-HTR (ours), $T$=1 | 3.489 |
| ANN-isomorphic (GELU), $T$=1 | 3.558 |
| ANN-pure: no InkCoder /no mem residual, $T$=1 | 3.646 |

*Table 4.* InkCoder ablations (validation CER, %).

| Variant | IAM | LAM | READ2016 |
|---|---|---|---|
| Full (InkCoder) | 3.5 | 2.3 | 4.2 |
| Repeat injection | 4.3 | 2.9 | 5.0 |
| Poisson coding | 4.7 | 3.1 | 5.3 |
| TTFS-style coding | 4.1 | 2.8 | 4.9 |
| Intensity-only proxy | 3.6 | 2.3 | 4.2 |
| Edge-only proxy | 3.8 | 2.4 | 4.5 |
| No contraction (fixed $\theta_t$) | 3.7 | 2.4 | 4.4 |

2.5 on LAM, and 3.9 on READ2016, without external language models and with greedy CTC decoding. Sweeping $T \in \{1, 2, 4\}$ shows that CER largely saturates by $T$=2; a full CER/WER breakdown is reported in Appendix Table 11. Throughout the $T$-sweep, we keep the preview-driven reduction hyperparameters fixed so that $\ell_b$ (hence token-mixing compute) remains comparable and changes in error primarily reflect temporal budget. In particular, increasing $T$ yields consistent WER reductions even when CER changes are small, indicating that extra timesteps mainly refine word level boundary/insertion errors rather than character substitutions.

### 4.2. Ablation Study

Unless stated otherwise, ablations use $T$=2 and the base training recipe. We isolate three aspects of budget alignment: (i) temporal entry (InkCoder) that enables meaningful small-$T$ refinement, (ii) token-length control that sets $\ell_b$ (reported via $r$) before deep mixing, and (iii) spiking activity patterns summarized by firing-rate and spike-volume statistics (Figure 4). Additional ablations of the spiking 2D encoder and the 1D mixer layout are reported in Appendix C.

To isolate the contribution of spiking dynamics under a strict timestep budget, we construct an ANN-isomorphic variant that preserves the architecture and training protocol but replaces spiking neurons with GELU (still $T$=1). On IAM validation, Spike-HTR reaches 3.489% CER versus 3.558% for the ANN-isomorphic control (Table 3), indicating a small advantage in this controlled IAM setting under an identical timestep budget. Removing both InkCoder and the membrane-residual bypass further degrades to 3.646%

CER, showing that these components provide measurable gains even at $T$=1.

InkCoder deterministically allocates a static image into a short temporal stream aligned with stroke evidence. Compared with common static-to-temporal baselines, InkCoder yields consistent gains across IAM, LAM, and READ2016 (Table 4). The proxy ablations clarify why: intensity cues provide a strong backbone, edge cues are complementary but weaker alone, and the gradual contraction over timesteps is important for sharpening selectivity without sacrificing coverage. Together, these results support the design choice that time in offline HTR should be used for structured refinement rather than redundancy or stochastic perturbation.

Token reduction controls the effective mixing length before the deep 1D mixer. Table 5 reports validation CER together with the measured reduction ratio $r$, where lower values indicate cheaper token mixing. Removing reduction ($r$=1) changes CER only marginally, while prune-only or merge-only variants underperform keep-and-merge, indicating that retaining key positions and collapsing long blank runs play complementary roles. Notably, keep-and-merge reduces the mixing length by about 32% with negligible CER change relative to no reduction, suggesting that most width positions are blank-redundant. Allowing gradients through the preview degrades CER at matched $r$, consistent with the preview becoming an auxiliary training signal rather than a pure budgeting mechanism. The minimum keep ratio

*Table 5.* Reduction ablations (validation CER, %). $r = \mathbb{E}[\ell_b/L_0]$; lower $r$ implies cheaper mixing.

| Variant | $r$ | IAM | LAM | READ2016 |
|---|---|---|---|---|
| Full (keep-and-merge) | 0.68 | 3.49 | 2.26 | 4.23 |
| No reduction ($L=L_0$) | 1.00 | 3.52 | 2.28 | 4.20 |
| Prune-only (no merge) | 0.73 | 3.51 | 2.27 | 4.21 |
| Merge-only (no prune) | 0.85 | 3.60 | 2.33 | 4.32 |
| Backprop through preview | 0.68 | 3.65 | 2.39 | 4.35 |

*Table 6.* Event sparsity statistics across all MultiStepLIF nodes, measured on 512 IAM samples ($d$=768, $T$=2).

| Modules | Mean | Median | p90 | p99 | Min | Max |
|---|---|---|---|---|---|---|
| 48 | 0.0379 | 0.0218 | 0.0911 | 0.1512 | 0.0044 | 0.1660 |

constrains $|K|$, whereas the reported $r$ is measured after merging; therefore $r$ can be slightly lower than the keep ratio when long blank runs collapse into short spans.

### 4.3. Activity and Energy Diagnostics

Spike-HTR matches accuracy to two budgets: token reduction shortens the effective sequence for deep 1D mixing, while spike activity governs the event volume of spiking computation. We therefore profile spike activity to pinpoint where events originate in an accurate checkpoint. We instrument all MultiStepLIF nodes and record emitted spike tensors during inference. For module $m$, let $S_m$ be total emitted spikes aggregated over timesteps and samples, and let $N_m$ be the number of spike elements per timestep. We report firing rate and spike-share as:

$$r_m = \frac{S_m}{T \, N_m}, \qquad s_m = \frac{S_m}{\sum_j S_j}. \qquad (21)$$

These metrics decouple activity intensity from event contribution: since $S_m \propto r_m N_m$, modules with small activation maps can fire more intensely yet contribute fewer total spikes. On 512 IAM line images using our base checkpoint, activity is globally sparse across 48 spiking modules (Table 6; Figure 4). While the highest firing rates appear in a small subset of 1D mixer outputs, stage aggregation reveals that total spike volume is dominated by early 2D encoder layers (Table 7), consistent with their much larger spatial activation maps.

This diagnosis suggests that further reducing event volume will most likely require earlier spatial compression or more spike-native early processing, rather than only refining the 1D mixer where activation maps are already compact. Full protocol details, module level breakdowns, and compute-only proxy analysis are provided in Appendix D.

*Table 7.* Stage level aggregation of event activity on 512 IAM samples ($d$=768, $T$=2). The 1D mixer exhibits higher activity intensity, while the 2D encoder dominates spike volume due to larger maps.

| Group | Mean firing rate | Median firing rate | Total spike-share |
|---|---|---|---|
| 2D encoder | 0.0176 | 0.0175 | 0.9665 |
| 1D mixer | 0.0581 | 0.0389 | 0.0335 |

## 5. Conclusion

We view offline HTR with SNNs as a problem of allocating limited temporal and sequence computation to sparse handwriting evidence: ink is spatially sparse, and CTC frame sequences are typically dominated by blanks. Spike-HTR uses two budgets, $T$ and $\ell_b$: InkCoder forms a short coarse-to-fine evidence stream, while the CTC-guided reducer shortens blank-heavy width sequences before deep mixing. The reducer is not specific to SNNs and may be reused in ANN-CTC recognizers, whereas InkCoder addresses the static-to-temporal entry problem specific to low-timestep spiking recognition. Beyond token reduction, activity-proportional efficiency will require more spike-native early layers and execution support for structured sparsity. Robustness under severe degradations, non-Latin scripts, and document-level layouts remains an important direction.

## Acknowledgments

This work was partly supported by Ningbo Key R&D Program (2025Z047) , Ningbo Major Application Demonstration Program (2025Z199) and Ningbo Youth Science and Technology Innovation Leading Talent Project (2024QL044).

## Impact Statement

This work aims to improve the efficiency of line level handwritten text recognition by aligning computation with stroke evidence and blank dominated CTC structure. Potential benefits include more compute aware digitization of historical documents and transcription workflows where runtime or hardware budgets are limited. Potential risks arise when efficient recognition is applied to sensitive handwritten records such as letters, forms, diaries, or archives. Deployment should require appropriate authorization, privacy protection, access control, and data governance. Because handwriting style, language, script, and scan quality vary widely, practitioners should evaluate target domain coverage and avoid relying on automated transcripts for high stakes decisions. Reported efficiency benefits should be interpreted as budget and activity diagnostics, with practical deployment gains depending on packing, sparse kernels, or event driven hardware.

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

## A. InkCoder

### A.1. Formulation

This section specifies the operators used to construct InkCoder evidence proxies and temporal gates. We use $X_{\text{gray}} \in [0,1]^{B \times 1 \times H \times W}$ for the grayscale image derived from $\mathbf{X}$. Quantiles used for evidence normalization are computed per sample over spatial positions.

We define three reusable operators. Let $\text{Quantile}(A, q)$ denote the per-sample spatial quantile of map $A$. Quantile normalization with inversion:

$$\mathcal{Q}(A; q_{\text{low}}, q_{\text{high}}) = 1 - \text{clip}\left( \frac{A - \text{Quantile}(A, q_{\text{low}})}{\text{Quantile}(A, q_{\text{high}}) - \text{Quantile}(A, q_{\text{low}}) + \epsilon}, 0, 1 \right). \tag{22}$$

A soft keep gate:

$$\mathcal{G}(A; \kappa, \tau) = \sigma(\kappa(A - \tau)). \tag{23}$$

A density suppressor:

$$\mathcal{D}(A; \kappa, \tau, \gamma) = \mathcal{G}(\text{BoxBlur}(A); \kappa, \tau)^{\gamma}. \tag{24}$$

The intensity proxy is obtained by smoothing and applying quantile normalization:

$$I_{\text{int}} = \mathcal{Q}(\text{BoxBlur}(X_{\text{gray}}); q_{\text{low}}, q_{\text{high}}). \tag{25}$$

For the edge proxy, let $S(\cdot)$ denote Sobel magnitude applied to a blurred image. We form a two-scale magnitude to recover faint strokes:

$$\begin{aligned} E^{(1)} &= S(\text{BoxBlur}(X_{\text{gray}})), \\ E^{(s)} &= \text{Up}_s\Big( S(\text{Down}_s(\text{BoxBlur}(X_{\text{gray}}))) \Big), \\ E &= \max\Big( E^{(1)}, E^{(s)} \Big), \end{aligned} \tag{26}$$

where $\text{Down}_s$ is average pooling by factor $s$ and $\text{Up}_s$ upsamples back to $(H, W)$. We normalize by a high quantile and apply two suppressors with distinct roles. First, cross-modal consistency keeps edges mainly where intensity also indicates ink:

$$\tilde{E} = \text{clip}\left( \frac{E}{\text{Quantile}(E, q_{\text{edge}}) + \epsilon}, 0, 1 \right), \qquad E_{\text{cm}} = \tilde{E} \cdot \mathcal{G}(I_{\text{int}}; \kappa_{\text{int}}, \tau_{\text{int}}). \tag{27}$$

Second, spatial coherence suppresses isolated responses by a density gate:

$$I_{\text{edge}} = E_{\text{cm}} \cdot \mathcal{G}(\text{BoxBlur}(E_{\text{cm}}); \kappa_{\text{e}}, \tau_{\text{e}}). \tag{28}$$

InkCoder allocates proxy emphasis over time through a step-dependent convex fusion:

$$D_t^{\text{raw}} = \text{clip}\big((1 - w_t)I_{\text{int}} + w_t I_{\text{edge}}, 0, 1\big), \qquad t = 1, \ldots, T. \tag{29}$$

To suppress speckle-like artifacts on fused evidence, we apply a density keep mask to each step:

$$D_t = D_t^{\text{raw}} \cdot \mathcal{D}(D_t^{\text{raw}}; \kappa_{\text{D}}, \tau_{\text{D}}, \gamma_{\text{D}}). \tag{30}$$

Gates are obtained by scheduled soft-thresholding:

$$g_t = \sigma(a_t(D_t - \theta_t)), \qquad t = 1, \ldots, T. \tag{31}$$

We parameterize schedules using normalized time $\lambda_t = \frac{t-1}{\max(T-1, 1)}$. The gate threshold follows a monotone raw schedule rather than a per-sample quantile:

$$\theta_t = \theta_{\min} + (\theta_{\max} - \theta_{\min})\lambda_t^{\gamma_\theta}, \qquad a_t = a_0(1 - \eta_a \lambda_t)s_\alpha + b_\alpha, \qquad w_t = \sigma\big(s_w(\lambda_t + b_w)\big). \tag{32}$$

The rise of $\theta_t$ admits broad stroke evidence early and tightens selectivity later, while $s_\alpha$ and $b_\alpha$ provide lightweight learned calibration of gate sharpness.

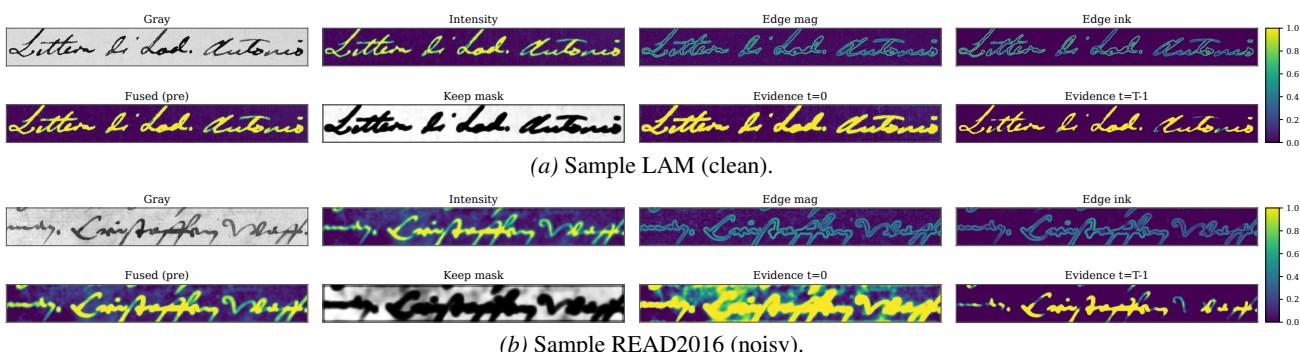

*(a)* Sample LAM (clean).

*(b)* Sample READ2016 (noisy).

*Figure 5.* InkCoder evidence proxies and suppression. Top row shows $X_{\text{gray}}$, the intensity proxy $I_{\text{int}}$, the normalized edge magnitude, and the edge proxy $I_{\text{edge}}$ after cross-modal and coherence suppression. Bottom row illustrates the density keep mask on fused evidence and the step-conditioned evidence $D_t$ at early and late steps.

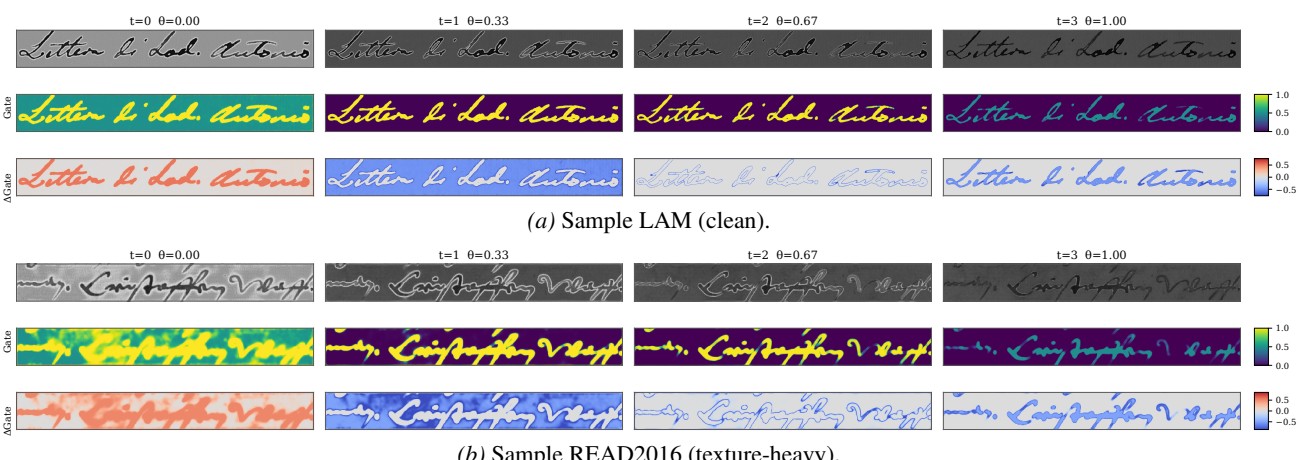

*(a)* Sample LAM (clean).

*(b)* Sample READ2016 (texture-heavy).

*Figure 6.* InkCoder temporal dynamics. Columns correspond to steps $t = 1, \ldots, T$ with thresholds $\theta_t$ shown in the headers. The top row visualizes a gated-input view for intuition, $\tilde{X}_t = X_{\text{gray}} \odot \left( \beta + (1-\beta)g_t \right)$; in the model, gates modulate the injected drive into the spiking pathway. The middle row shows $g_t$. The bottom row shows temporal novelty $\Delta g_t = g_t - g_{t-1}$ for $t \geq 2$ (with $\Delta g_1 = g_1$), highlighting where evidence is removed or reinforced as selectivity increases.

## A.2. Diagnostics

We visualize (i) the evidence proxies and suppressors, and (ii) gate dynamics across timesteps (Figures 5 and 6). All maps are produced by the same operators and schedules as used at inference time. We render them at the model's normalized aspect ratio (upsampled only for readability, preserving geometry).

## B. Experimental Setup

This section summarizes the unified protocol used across all experiments and reports the base configuration for reproducibility. Unless otherwise stated, experiments vary one budget at a time while keeping the remaining configuration fixed.

### B.1. Evaluation Protocol and $T$-sweep

We train each model from scratch on the target dataset only (no external pretraining, synthetic data, language models, or lexicons) and decode with greedy CTC. We select the checkpoint with the lowest validation CER (computed with EMA weights) and report the corresponding test metrics. In the main scaling study, we sweep $T \in \{1, 2, 4\}$ while keeping the preview-driven reduction hyperparameters fixed (including $\tau, \gamma, k$ and the InkCoder schedules), so changes in CER primarily reflect temporal budget rather than a modified reduction rule.

## B.2. Datasets

We use the line level splits summarized in Table 2. IAM (Marti & Bunke, 2002) uses a writer-disjoint split, LAM (Cascianelli et al., 2022) isolates long-term drift of a single writer, and READ2016 (Sanchez et al., 2016) contains historical degradations and variable orthography. The character vocabulary is built from the training split; transcripts at train/validation/test time are normalized consistently using Unicode NFKC, collapsing consecutive whitespace, and removing the U+00AC artifact symbol. Samples with empty transcripts are filtered. To avoid dataset-specific priors, we do not apply lexicon constraints or language-model decoding at test time.

## B.3. Model and Hyperparameters

Tables 8 and 9 record the base configuration used unless otherwise stated. Module definitions are given in Section 3; here we list concrete dimensions and the default settings for the two budgets. The spiking horizon is fixed per run (base: $T = 2$); for the $T$-sweep we only change $T$ and keep all other hyperparameters unchanged. Blank-guided reduction uses the same preview-driven keep-and-merge rule for all runs, with fixed thresholds $(\tau, \gamma, k)$ (Table 8). Temporal injection uses a non-zero baseline $\beta$ (Table 8) to avoid vanishing drive on low-evidence samples, and dual-resolution downsampling uses a learnable average/max mixture initialized to a stable value.

*Table 8.* Notation and primary hyperparameters of the base Spike-HTR configuration.

| Symbol | Meaning / Value in our base setup |
|---|---|
| $B$ | batch size per GPU (train: $4$) |
| $H$ | normalized image height ($64$) |
| $W$ | image width after preprocessing (max $512$; right-padded) |
| $T$ | SNN simulation timesteps (base: $2$) |
| $T_{\max}$ | temporal fusion buffer cap for shape safety (base: $4$) |
| $d$ | embedding dimension ($768$) |
| $h$ | number of heads in the 1D mixer ($12$) |
| $\beta$ | temporal injection baseline (base: $0.35$) |
| $p_{\mathrm{drop}}$ | dropout rate ($0.10$) |
| $p_{\mathrm{sd}}$ | stochastic depth rate ($0.20$; linearly ramped across blocks) |
| $c_1, c_2$ | 2D stage widths: $c_1 = d/4 = 192$, $c_2 = d/2 = 384$ |
| $D_{\mathrm{enc}}$ | 2D encoder depth: $(1, 1, 5)$ (total depth $7$) |
| $N_{\mathrm{mix}}$ | 1D mixer depth ($6$ blocks) |
| $L_0$ | pre-reduction length ($L_0 = W/4 \leq 128$) |
| $\ell_b$ | post-reduction valid length for sample $b$ (dynamic; $\ell_b \leq L_0$) |
| $L$ | padded length in a batch ($L = \max_b \ell_b$) |
| $\tau$ | blank keep threshold (keep if $p_{\mathrm{blank}} < \tau$; base: $0.88$) |
| $\gamma$ | minimum keep ratio (base: $0.70$) |
| $k$ | merge span size cap (base: $3$) |
| $V$ | vocabulary size (dataset-dependent; includes blank with id $= 0$) |

The base model contains 76.66M trainable parameters; the 1D mixer accounts for approximately $59.6\%$ of the total. Table 9 provides a stage-wise view of tensor shapes and where token-length reduction is applied.

## B.4. Training

We train with an epoch-based protocol and select checkpoints by validation CER computed using EMA weights (Section B.1). We use AdamW with a cosine learning-rate schedule and keep the training recipe fixed across ablations; in particular, the $T$-sweep uses the same optimizer, schedule, and number of epochs for all $T$.

Input images are normalized to a fixed height $H = 64$ and padded on the right to a maximum width $W \leq 512$. CTC input lengths are computed from the post-reduction valid lengths $\{\ell_b\}$ produced by the reducer. The reducer enforces a deployment-valid minimum keep ratio $\gamma$; during training, if a rare sample violates feasibility ($\ell_b < U_b$), we relax reduction for that sample to ensure $\ell_b \geq U_b$ rather than clamping lengths. The software and hardware stacks used for the single-GPU and multi-GPU runs are reported in Table 10.

*Table 9.* Stage-wise architecture and tensor shapes in Spike-HTR. Token length is reduced before the 1D mixer.

| Stage | Blocks (Configuration Details) | Output Shape |
|---|---|---|
| **Static stem to temporal injection** | | |
| Input | RGB image normalized to $[0, 1]$ | $[B, 3, H, W]$ |
| Stem (static) | Conv2d$(3 \rightarrow c_1)$ + GroupNorm + SiLU | $[B, c_1, H, W]$ |
| Temporal injection | Broadcast stem to $T$ and modulate via time-indexed gates with baseline $\beta$ | $[T, B, c_1, H, W]$ |
| **Multi-step spiking 2D encoder** | | |
| Stage 1 | Multi-step spiking conv $(c_1 \rightarrow c_1)$ + ConvMix2d $\times 1$ | $[T, B, c_1, H, W]$ |
| Stage 2 | Multi-step spiking conv $(c_1 \rightarrow c_2, s{=}2)$ + ConvMix2d $\times 1$ | $[T, B, c_2, H/2, W/2]$ |
| Stage 3 | Multi-step spiking conv $(c_2 \rightarrow d, s{=}2)$ + ConvMix2d $\times 5$ | $[T, B, d, H/4, W/4]$ |
| Fusion | Dual-resolution fusion with gated injection; downsample mixes avg/max | $[T, B, d, H/4, W/4]$ |
| **Tokenization and dynamic reduction** | | |
| Pooling | Attention-gated height pooling with channel-wise max nudging | $[T, B, d, L_0]$ |
| Positional | Absolute positional embedding + convolutional positional encoding ($k = 7$) | $[T, B, d, L_0]$ |
| Reduction | Stop-gradient shared-classifier preview, then keep-and-merge (cap $k$, keep ratio $\gamma$) | $[T, B, d, L]$ |
| **1D sequence mixer and heads** | | |
| Mixer blocks ($N{=}6$) | Hybrid mixing with linear attention, QK mixing, and Spiking LKBlock | $[T, B, d, L]$ |
| Main head | Temporal fusion, LayerNorm, Linear$(d \rightarrow V)$ | $[B, L, V]$ |
| Aux head | Shallow tap, aligned prediction head | $[B, L, V]$ |

*Table 10.* Training configuration of the base model.

| Item | Setting |
|---|---|
| **Data and text processing** | |
| Splits | Train/val/test follow the official protocols; charset derived from the training split; samples with empty transcripts filtered |
| Normalization | Unicode NFKC; collapse consecutive whitespace; remove U+00AC artifact symbol |
| Input geometry | Resize to $H = 64$ with bicubic interpolation; width right-padded to max $W \leq 512$; aspect-ratio-preserved downscaling if needed |
| DataLoader | Workers $= 4$; pinned memory enabled; prefetching enabled |
| **Objective and decoding** | |
| Main loss | CTC loss with blank id $= 0$ and zero_infinity enabled; computed in fp32 |
| Auxiliary loss | Auxiliary CTC from shallow tap with weight 0.2 |
| Consistency | Symmetric KL between main and auxiliary predictions with weight 0.05 and temperature 1.0 |
| Decoding | Greedy CTC decoding; consistent text normalization applied; no external language model, lexicon, or synthetic data |
| Feasibility | Minimum keep ratio $\gamma$ enforced for deployment. If a rare training sample violates feasibility, namely $\ell_b < U_b$, reduction is relaxed for that sample so that $\ell_b \geq U_b$ |
| **Optimization** | |
| Optimizer | AdamW; learning rate $3 \times 10^{-4}$; weight decay 0.05; gradient clipping norm 1.0 |
| Schedule | Cosine schedule; warmup 5 epochs; minimum learning rate $1 \times 10^{-6}$; total 210 epochs |
| EMA | Enabled with decay 0.9998; updated at every optimization step |
| **Regularization and augmentation** | |
| Dropout / stochastic depth | Dropout rate 0.10; stochastic depth rate 0.20, linearly scheduled across blocks |
| Augmentations | Affine transform; width stretch; noise or sharpen; stroke jitter; inversion disabled by default |
| **Implementation** | |
| Precision | Native AMP and TF32 enabled; fp32 enforced for CTC loss, normalization, and temporal reductions |
| Software | PyTorch 2.6/2.9 with CUDA 12.4/13.0 |
| Hardware | Training runs used one NVIDIA RTX 6000D GPU, eight NVIDIA GeForce RTX 4090 D GPUs, or NVIDIA Tesla V100-SXM2-32GB GPUs |

## C. Additional Ablations

This section complements the main ablation study in Section 4 by (i) quantifying the effect of temporal budget $T$ under a fixed reduction policy, (ii) reporting a model-scale ablation across capacity tiers, and (iii) isolating two architectural choices that affect handwriting morphology and long-range sequence modeling: preservation mechanisms in the spiking 2D encoder and the operator composition of the 1D mixer.

### C.1. Temporal Budget Sweep

Table 11 extends the main temporal-budget study by reporting CER/WER on all three benchmarks under the same fixed reduction rule.

*Table 11.* Effect of temporal budget $T$ (CER/WER in %).

| $T$ | IAM | | LAM | | READ2016 | |
|---|---|---|---|---|---|---|
| | Val | Test | Val | Test | Val | Test |
| 1 | 3.5/13.9 | 5.5/19.4 | 2.3/8.6 | 2.6/9.7 | 4.2/21.6 | 4.1/20.2 |
| 2 | 3.5/13.8 | 5.4/18.9 | 2.3/8.3 | 2.5/9.4 | 4.2/21.3 | **3.9**/20.0 |
| 4 | 3.5/**13.4** | 5.4/**18.8** | 2.3/**7.6** | 2.5/**8.5** | **4.1/21.1** | 4.0/**19.7** |

Across datasets, increasing $T$ yields small but consistent WER reductions (IAM test $19.4 \to 18.8$, LAM $9.7 \to 8.5$, READ2016 $20.2 \to 19.7$), while CER is largely saturated. This suggests that additional temporal steps mainly reduce word level insertion/deletion and boundary errors (which disproportionately affect WER) rather than changing overall character substitution rates. Gains diminish beyond $T=2$; on READ2016, $T=4$ slightly increases CER ($3.9 \to 4.0$), consistent with iterative refinement amplifying low-SNR clutter. While the 1D mixer cost is unchanged under fixed reduction, the spiking backbone scales roughly with $T$ (see Appendix D), so we use $T=2$ as the default trade-off and recommend $T=4$ when WER is prioritized.

### C.2. Model Scale Ablation on IAM

We study how Spike-HTR scales with model capacity and whether larger models can better exploit longer spiking horizons. We evaluate three capacity tiers (tiny/small/medium) on IAM and sweep $T \in \{1, 2, 4\}$ within each tier. Within a tier, training and reduction settings are fixed; across tiers, we adopt the default tier-specific settings used in our training code for stability. We select checkpoints by IAM validation CER (EMA) and report the corresponding validation CER/WER under greedy CTC. Table 12 lists the capacity-tier configurations, and Table 13 reports the corresponding IAM validation CER/WER under different temporal budgets.

*Table 12.* Configurations for the IAM model-scale ablation. "LA/QK/LK" denote LinearAttnBlock1D, QKMixBlock1D, and SpikingLK-Block1D in the 1D mixer. $k_{LK}$ and $e_{LK}$ denote the kernel size and channel expansion ratio of the large-kernel convolutional mixer, respectively. $\gamma$ is the minimum keep ratio, $\tau$ is the blank threshold, and $k$ is the maximum merge span. "Aux/Cons." indicates whether the auxiliary CTC branch and masked symmetric-KL consistency are enabled (Section 3.4).

| Model | Params (M) | $d$ | $N_{\mathrm{mix}}$ | Mixer layout | MLP | $k_{\mathrm{LK}}$ | $e_{\mathrm{LK}}$ | $\gamma$ | $\tau$ | $k$ | Aux/Cons. |
|---|---|---|---|---|---|---|---|---|---|---|---|
| tiny | 8.5 | 384 | 4 | LA–QK–LK–LA | 2.5 | 15 | 1.00 | 0.75 | 0.90 | 3 | – |
| small | 18.6 | 512 | 4 | LA–QK–LK–LA | 3.0 | 21 | 1.25 | 0.70 | 0.88 | 3 | ✓ |
| medium | 39.9 | 640 | 5 | LA–QK×2–LK–LA | 3.5 | 27 | 1.60 | 0.70 | 0.88 | 3 | ✓ |

*Table 13.* IAM validation CER/WER (%; greedy CTC) for model scale × temporal budget. Best CER within each row is highlighted.

| Model | $T=1$ | $T=2$ | $T=4$ |
|---|---|---|---|
| tiny | 4.38/16.33 | **4.31/16.42** | 4.32/16.47 |
| small | 3.95/15.22 | **3.85/14.70** | 3.86/14.61 |
| medium | 3.73/14.18 | 3.71/14.10 | **3.65/13.80** |

Model scaling yields consistent gains: from tiny to medium, validation CER drops by 0.66 points at $T=2$ ($4.31 \to 3.71$)

while WER drops by 2.32 points (16.42 → 14.10). Increasing $T$ helps the larger tiers (small/medium) more than tiny: medium benefits from longer horizons in both CER and WER, whereas tiny shows saturated (and slightly worsening) WER. This pattern is consistent with longer temporal horizons amplifying refinement capacity only when the 1D mixer is sufficiently expressive and the preview remains well-calibrated; enabling the auxiliary CTC + cross-depth consistency in the larger tiers further stabilizes this calibration.

### C.3. 2D Encoder Preservation Mechanisms

The spiking 2D encoder includes an analog bypass (Eq. 8) and dual-resolution fusion (Eq. 9) to counteract low-pass integration and preserve fine stroke topology. Removing the bypass makes the model more brittle on READ2016, consistent with losing weak-but-consistent grayscale evidence that helps resolve degraded strokes. Removing dual-resolution fusion degrades all datasets and particularly hurts READ2016, where fine stroke fragments can be confused with background texture if higher-resolution cues are not reinjected. The learnable avg/max mixture yields modest but consistent gains, reflecting a data-dependent balance between background stabilization and thin-stroke preservation. Table 14 reports the validation CER changes caused by removing or simplifying these preservation mechanisms.

*Table 14.* 2D encoder ablations (validation CER, %).

| Variant | IAM | LAM | READ2016 |
|---|---|---|---|
| Full (spiking 2D encoder) | 3.4 | 2.3 | 4.2 |
| w/o analog bypass ($\alpha{=}0$) | 3.8 | 2.4 | 4.7 |
| w/o dual-resolution fusion | 3.7 | 2.3 | 4.4 |
| Avg-only downsample | 3.6 | 2.3 | 4.5 |
| Max-only downsample | 3.7 | 2.3 | 4.4 |

### C.4. 1D Mixer Operator Composition

Each block composes three complementary token-mixing operators (details and ablations in Appendix C): (i) linear attention for stable global aggregation, (ii) lightweight QK-style mixing for content gating, and (iii) a spike-friendly large-kernel convolutional mixer (SpikingLKBlock) for midrange continuity, followed by a spiking MLP. Table 15 compares the hybrid mixer with single-operator mixer variants.

*Table 15.* Mixer layout ablations (validation CER, %).

| Layout | IAM | LAM | READ |
|---|---|---|---|
| Auto (LA→QK→LK hybrid) | 3.4 | 2.3 | 4.2 |
| All LinearAttn | 3.55 | 2.28 | 4.20 |
| All QKMix | 3.63 | 2.32 | 4.28 |
| All LKBlock | 3.58 | 2.27 | 4.22 |

## D. Energy Analysis

This section reports reproducible event-sparsity measurements and a compute-only arithmetic proxy for Spike-HTR. We focus on (i) spike activity statistics collected from instrumented spiking modules and (ii) an operation-count proxy that isolates arithmetic trends under event-driven accounting.

### D.1. Event-sparsity Profiling Protocol

We instrument all MultiStepLIF nodes and record the emitted binary spike tensors during inference. Unless stated otherwise, all statistics are computed on 512 IAM samples using the checkpoint with $d{=}768$ and $T{=}2$. For each spiking module $m$, we report firing rate and spike-share defined in Eq. (21). The main paper reports aggregate distribution statistics (Table 6), the distribution visualization, and stage aggregation (Table 7). Here we provide finer module level breakdowns to identify dominant contributors and to support reproducibility.

## D.2. Module Breakdown and Stage Dominance

Firing rate (intensity) and spike-share (volume) can diverge. High firing rates are typically observed in a small subset of sequence-mixing outputs, while dominant spike volume can arise from early 2D layers due to larger activation maps. To make this explicit, Table 16 aggregates activity by stage group, and Table 17 lists representative modules that dominate by volume and by intensity.

*Table 16.* Group level spike dominance on the IAM profiling set ($d$=768, $T$=2). The 1D mixer shows higher activity intensity, while the 2D encoder dominates spike volume due to larger activation maps.

| Group | #Modules | Mean $r_m$ | Median $r_m$ | $\sum s_m$ |
|---|---|---|---|---|
| 2D encoder | 24 | 0.0176 | 0.0175 | 0.9665 |
| 1D sequence mixer | 24 | 0.0581 | 0.0389 | 0.0335 |

*Table 17.* Top modules by spike-share (volume) and by firing rate (intensity). Enc-S$k$-B$j$-X denotes encoder stage $k$, block $j$, component X; Seq-$i$-X denotes the $i$-th 1D mixer block component X.

| | Top by spike-share | | | Top by firing rate | |
|---|---|---|---|---|---|
| Module | $s_m$ | $r_m$ | Module | $r_m$ | $s_m$ |
| Enc-S1-B0-PW1 | 22.8% | 0.021 | Seq-4-LK-OUT | 0.166 | 0.3% |
| Enc-S1-B0-PW2 | 8.0% | 0.029 | Seq-5-MLP-FC2 | 0.135 | 0.3% |
| Enc-S1-B0-DW | 7.0% | 0.026 | Seq-4-MLP-FC2 | 0.126 | 0.2% |
| Enc-S2-B0-PW1 | 6.6% | 0.012 | Seq-3-LK-OUT | 0.122 | 0.2% |
| Enc-S3-B4-PW1 | 6.4% | 0.023 | Seq-0-MLP-FC2 | 0.094 | 0.2% |

## D.3. Compute-only Arithmetic Proxy

We use a compute-only proxy that compares arithmetic energy under fixed per-operation costs. This proxy ignores memory traffic and data movement and therefore should not be interpreted as a GPU/TPU energy model; it is included only as a relative arithmetic accounting for scenarios where operations become eligible for accumulation-style execution.

Following common 45 nm-style estimates, a 32-bit floating multiply costs $3.7\,\mathrm{pJ}$ and an add costs $0.9\,\mathrm{pJ}$, hence

$$E_{\mathrm{MAC}} = 4.6\,\mathrm{pJ}, \qquad E_{\mathrm{AC}} = 0.9\,\mathrm{pJ}, \tag{33}$$

and $\rho = E_{\mathrm{MAC}}/E_{\mathrm{AC}}$.

Let $N_{\mathrm{MAC}}^{\mathrm{dense}}$ denote the dense MAC count of the same architecture under dense arithmetic. We partition arithmetic into dense MACs that remain dense and event-driven ACs that scale with measured activity:

$$E_{\mathrm{dense}} = N_{\mathrm{MAC}}^{\mathrm{dense}} E_{\mathrm{MAC}}, \qquad E_{\mathrm{hyb}} = N_{\mathrm{MAC}}^{\mathrm{hyb}} E_{\mathrm{MAC}} + N_{\mathrm{AC}}^{\mathrm{hyb}} E_{\mathrm{AC}}. \tag{34}$$

The normalized ratio is

$$\frac{E_{\mathrm{hyb}}}{E_{\mathrm{dense}}} = \frac{N_{\mathrm{MAC}}^{\mathrm{hyb}} + \frac{1}{\rho} N_{\mathrm{AC}}^{\mathrm{hyb}}}{N_{\mathrm{MAC}}^{\mathrm{dense}}}. \tag{35}$$

Eligibility is conservative: a layer is counted as event-driven only if its input is sufficiently spike-like; otherwise it is counted as dense MAC.

## D.4. Proxy Results and Sensitivity

Table 18 summarizes the dense reference, conservative default estimate, pure-spike variant, and idealized upper bound under the compute-only proxy. The default setting follows the current architecture and the conservative eligibility rule above. The pure-spike setting removes the analog bypass and therefore increases the share of arithmetic that can be counted as activity-scaled, but it should be read together with the accuracy ablation in Table 14, where removing the bypass hurts recognition. The ideal setting is not an implementation result. It is an optimistic upper bound that assumes all non-stem

*Table 18.* Compute-only energy proxy comparison on 512 IAM samples ($d{=}768$, $T{=}2$). Counts are per-sample averages. The ratio is reported under $\rho = E_{\mathrm{MAC}}/E_{\mathrm{AC}}$ and is also swept for $\rho \in \{5, 10, 20, 30\}$.

| Setting | $N_{\mathrm{MAC}}^{\mathrm{dense}}$ | $N_{\mathrm{MAC}}^{\mathrm{hyb}}$ | $N_{\mathrm{AC}}^{\mathrm{hyb}}$ | $\frac{E_{\mathrm{hyb}}}{E_{\mathrm{dense}}}$ ($\rho{=}5.11$) | Eligible share | Notes |
|---|---|---|---|---|---|---|
| Default | $1.952 \times 10^{11}$ | $1.250 \times 10^{11}$ | $1.517 \times 10^{9}$ | 0.642 | 0.359 | Base model |
| Pure-spike | $1.953 \times 10^{11}$ | $1.021 \times 10^{11}$ | $1.945 \times 10^{9}$ | 0.525 | 0.477 | Analog bypass disabled |
| Ideal | $1.953 \times 10^{11}$ | $1.699 \times 10^{8}$ | $4.07 \times 10^{9}$ | 0.00495 | 1.000 | All non-stem layers event-driven |

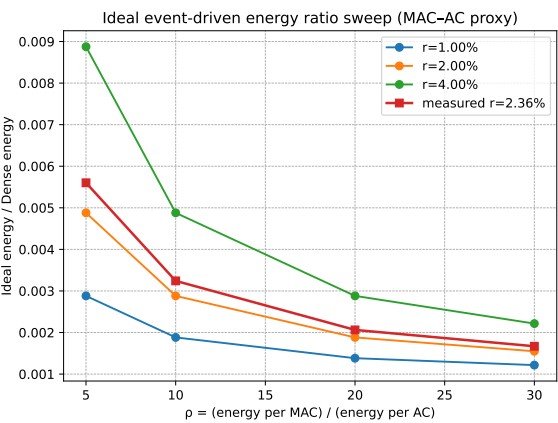

*Figure 7.* Idealized compute-only energy ratio under MAC–AC cost sweeps.

arithmetic becomes event-driven while the stem remains dense. Let $r$ denote an element-weighted global average firing rate measured from recorded spikes. Then

$$N_{\mathrm{MAC}}^{\mathrm{ideal}} = N_{\mathrm{MAC}}^{\mathrm{stem}}, \qquad N_{\mathrm{AC}}^{\mathrm{ideal}} = r\big(N_{\mathrm{MAC}}^{\mathrm{dense}} - N_{\mathrm{MAC}}^{\mathrm{stem}}\big). \tag{36}$$

We sweep $\rho \in \{5, 10, 20, 30\}$ to contextualize sensitivity to the assumed MAC–AC gap.

## E. Discussion

Spike-HTR suggests that offline HTR with SNNs is not merely a problem of backbone design, but also a problem of budget allocation. The first budget is temporal: a static line image should provide useful evidence over a small number of spiking steps rather than simply replaying the same frame. InkCoder instantiates this idea with a deterministic coarse-to-fine stream, where early steps preserve broad stroke support and later steps sharpen boundary-aligned evidence. The second budget is sequential: after height pooling, many CTC frames are blank-dominated, yet short blank intervals and uncertain positions can still be needed to separate neighboring or repeated characters. The CTC-aware reducer therefore shortens the width-axis stream before the deep 1D mixer while preserving likely non-blank positions, uncertain positions, and short separator regions.

These two components have different scopes. The length reducer is not specific to spiking models. It only requires a CTC head that provides blank posteriors and uncertainty estimates, and can therefore be viewed as a potentially reusable CTC-aligned length-control mechanism for both SNN and ANN recognizers. In contrast, InkCoder targets a spiking-specific difficulty: offline HTR provides static images, whereas SNN computation unfolds over timesteps. Under a strict small timestep budget, the goal is not to create a longer temporal simulation, but to make each timestep carry distinct and progressively refined handwriting evidence. Accordingly, the matched ANN control serves as a budget-matched reference point rather than a leaderboard comparison. The most directly reusable efficiency mechanism is the CTC-aware reducer, while the spiking-specific contribution lies in short-horizon temporal entry and the resulting low-activity operating regime.

On current dense accelerators, sparse internal firing alone does not determine wall-clock behavior, because the early visual pathway, spatial convolutions, normalization, memory movement, and padded batches are still executed largely by dense

kernels. The immediate algorithmic benefit therefore comes from reducing the width-axis length before the deep mixer. At the same time, the firing profile reveals where the spiking pathway already operates with low event volume and where dense or analog components still limit activity-proportional execution. This motivates a more spike-native treatment of the early visual interface, especially for converting static handwriting into temporally structured stroke evidence before deep sequence modeling.

Several limitations remain. First, the early visual pathway remains hybrid, including a dense stem and an analog shortcut used to preserve thin-stroke topology. These choices improve recognition robustness, but they reduce how spike-native the current implementation is. Second, InkCoder encodes handwriting priors through intensity, edge, and coherence cues. This makes the temporal entry stable under small timesteps, but the same cues may be distorted by extreme blur, bleed-through, textured backgrounds, unusual writing materials, or scripts with substantially different stroke statistics. In such cases, overly aggressive gating may suppress faint strokes, while overly permissive gating may increase activity and weaken the small timestep advantage. Third, the reducer produces variable-length sequences. Without length bucketing or packed execution, padding can hide part of the algorithmic benefit of reducing the sequence length. Finally, our experiments focus on line level Latin-script benchmarks, so additional validation is needed before extending the conclusions to non-Latin scripts, complex layouts, or document level recognition.

These limitations point to several directions for future work. A natural next step is to make the temporal entry more adaptive to degradation patterns and script dependent stroke statistics, while preserving the stability of the current coarse to fine design, for example through constrained monotone schedules or lightweight mixtures over evidence channels. For event-driven deployment, realizing activity proportional benefits will require more spike native early layers and execution support that can exploit structured sparsity rather than only measuring it in the representation. Broader validation on non-Latin scripts, more severe degradations, and document level layouts is also needed before treating the intensity, edge, and coherence priors used here as generally applicable.

