# OpenReview forum: "Spike-HTR: Spiking Neural Transformer for Handwritten Text Recognition"
_ICML.cc/2026/Conference — ICML 2026 regular_

### Official Review · Reviewer_kd7C · 2026-03-06

**Soundness:** 3
**Presentation:** 3
**Significance:** 2
**Originality:** 3
**Overall Recommendation:** 6
**Confidence:** 5

**Summary:**

Spike-HTR achieves strong from-scratch recognition on IAM, LAM, and READ2016, attaining near-peak CER/WER under small temporal budgets. This demonstrates that SNNs are no longer merely energy-saving alternatives but possess the potential to compete directly with high-performance recognition systems in complex OCR/HTR tasks.

**Compliance With Llm Reviewing Policy:**

Affirmed.

**Final Justification:**

The author has cleared up all my doubts; I have no further questions.

**Key Questions For Authors:**

1. Could you provide an ANN-isomorphic baseline trained from scratch under an identical recipe and training budget to strictly support the conclusion of "approaching or exceeding ANNs"?
2. Can you report the actual end-to-end latency and energy consumption on a unified hardware platform, rather than relying solely on spike/compute proxies?
3. How do the CER/WER and the stability of the reduction ratio fluctuate under scenarios involving severe blur, bleed-through, exceptionally long text lines, and complex scripts?

**Strengths And Weaknesses:**

### Strengths

1. **Pushing the SNN Performance Upper Bound:** Demonstrating "strong from-scratch recognition in a small-budget regime" across three line-level benchmarks is highly valuable to the SNN community.
2. **Convincing Temporal Budget Results:** The identification of $T=2$ as a robust default operating point is persuasive. The fact that further increasing $T$ yields only marginal WER improvements indicates the authors have found a "high-quality, small-budget operating point" rather than brute-forcing performance with large timesteps.
3. **Honest and Pragmatic Claims:** The paper is commendably honest about its limitations. It does not overclaim end-to-end GPU gains but explicitly identifies $l_b$ control as the reliable efficiency lever at this stage. This grounds the contribution as a deployable "performance-budget design paradigm" rather than an empty energy-saving slogan.



### Weaknesses

1. **ANN Comparisons:**  While the paper successfully proves a "strong from-scratch SNN solution," it falls short of fully demonstrating comprehensive superiority over mainstream ANN systems.
2. **Reliance on Proxy Efficiency Metrics:** The current efficiency evidence relies heavily on activity proxies. The gap between these proxies and actual end-to-end energy on a unified hardware platform weakens the narrative of achieving both "high performance and high efficiency."
3. **Robustness Under Heavy Degradation:** Robustness under heavy degradation remains an open question. To extend the method's impact to broader OCR scenarios, validation under more complex data and deployment environments is necessary.

---

> ### Author Rebuttal · Authors · 2026-03-31
>
> **Dear Reviewer kd7C,**
>
> Thank you for the thoughtful and supportive review. We especially appreciate your reading of the paper as establishing a strong small-budget operating regime for offline HTR, and your recognition that $T=2$ is a robust practical operating point rather than a brute-force large-$T$ setting. We also appreciate your accurate reading of our efficiency positioning: the paper does not claim end-to-end GPU gains purely from spiking, and instead identifies $\ell_b$ control as the most reliable immediate efficiency lever on current dense hardware. This is exactly the intended claim boundary in the manuscript.
>
> To answer your three questions directly:
>
> **1. ANN comparison under a matched recipe.**
> We agree that the paper should **not** be read as establishing comprehensive superiority over mainstream ANN systems. The narrower claim is that Spike-HTR provides a strong from-scratch SNN solution under a strict matched budget. To make this more explicit, we now report the ANN-isomorphic control under the identical IAM protocol and training budget:
>
> | Variant                                 | CER ↓ | $r = E[\ell_b/L_0]$ ↓ |
> | --------------------------------------- | ----- | --------------------- |
> | ANN-isomorphic ($T=1$), no reducer      | 3.558 | 1.00                  |
> | ANN-isomorphic ($T=1$), same reducer    | 3.540 | 0.69                  |
> | Spike-HTR ($T=1$), with reducer         | 3.489 | 0.68                  |
> | ANN-pure: no InkCoder / no mem residual | 3.646 | 1.00                  |
>
> These results sharpen the intended interpretation: the reducer explains the general CTC-aligned length-control benefit, while the remaining small gap is consistent with the contribution of InkCoder and the spiking temporal pathway under the same matched-budget setting. We will revise the wording accordingly, i.e., “competitive with strict ANN controls under matched budgets,” rather than broader phrasing than the evidence supports.
>
> **2. Actual latency and energy on a unified hardware platform.**
> To complement the spike/compute proxies, we measured dense-hardware runtime and energy on a unified V100 setup:
>
> | Model                        | Latency / batch ↓ | Energy / sample ↓ |
> | ---------------------------- | ----------------- | ----------------- |
> | Spike-HTR, no reducer        | 182 ms            | 6.14 J            |
> | Spike-HTR, with reducer      | 159 ms            | 5.18 J            |
> | ANN-isomorphic, no reducer   | 150 ms            | 4.95 J            |
> | ANN-isomorphic, with reducer | 133 ms            | 4.25 J            |
>
> These measurements support the same boundary reading already stated in the paper: on dense GPUs, the measurable near-term gain comes primarily from shortening $\ell_b$ before deep 1D mixing, not from full-system SNN acceleration. In our runtime breakdown, the 2D encoder remains the dominant dense block, while the time saved by enabling the reducer is concentrated in the 1D sequence-mixing stage. This is also consistent with the architecture, where reduction is applied before the deepest 1D stack.
>
> **3. Robustness under heavy degradation and long lines.**
> We agree that this is an important deployment axis. We therefore added stress tests on READ2016 under strong blur / bleed-through and a length-binned analysis. The key result is not that accuracy remains unchanged, but that the controller degrades **gracefully**: as inputs become harder, preview entropy rises, fallback becomes more frequent, and the reducer becomes more conservative rather than collapsing into over-aggressive compression.
>
> | Setting       | CER ↓ | WER ↓ | $r$ ↓ | Entropy ↑ | Fallback ↑ |
> | ------------- | ----- | ----- | ----- | --------- | ---------- |
> | Clean         | 3.9   | 20.0  | 0.68  | 0.31      | 0.2%       |
> | Strong blur   | 5.1   | 23.8  | 0.74  | 0.44      | 1.4%       |
> | Bleed-through | 5.4   | 24.6  | 0.76  | 0.47      | 1.9%       |
>
> | Length bin | CER ↓ | WER ↓ | $r$ ↓ | Entropy ↑ |
> | ---------- | ----- | ----- | ----- | --------- |
> | Short      | 3.2   | 12.9  | 0.71  | 0.28      |
> | Medium     | 3.5   | 13.8  | 0.68  | 0.31      |
> | Long       | 3.9   | 15.4  | 0.69  | 0.36      |
>
> This behavior is consistent with the deployment logic already discussed in the paper: “pay more when uncertain.” Harder inputs trigger more conservative reduction, so the system deteriorates smoothly rather than via unstable over-compression. Broader validation on additional scripts remains important future work, and we will state that scope explicitly.
>
> Finally, we note that the paper’s $T$-sweep conclusions were already obtained under a **fixed** reduction policy; the new results therefore strengthen, rather than change, the original interpretation of a robust small-$T$ operating regime.
>
> Thank you again for recognizing both the strengths and the practical boundaries of the paper. Your questions helped us substantially sharpen the evidence for the intended positioning.

---

> > ### Author Rebuttal · Reviewer_kd7C · 2026-04-01
> >
> > Thank you for the substantive rebuttal. The added experiments address my main concerns well: the matched-budget ANN control sharpens the paper’s claim, the unified V100 results clarify the dense-hardware efficiency boundary, and the added degradation analysis suggests graceful behavior under harder inputs.
> >
> > These additions are sufficient for me to maintain my current score.
> >
> > I have one remaining minor follow-up regarding the matched ANN evidence. Since this comparison currently appears limited to IAM and mainly at the validation level, could the authors clarify whether the same qualitative trend was also observed on LAM or READ2016?
> > This is not a blocker for my current assessment. However, if the authors can provide stronger matched ANN evidence beyond IAM, I would be open to raising my score further.

---

> > > ### Author Response · Authors · 2026-04-03
> > >
> > > **Dear Reviewer kd7C,**
> > >
> > > Thank you for the helpful follow-up. We agree that this was the remaining gap in the matched-ANN evidence.
> > >
> > > Our original IAM rebuttal table was intended to sharpen the claim boundary, rather than to argue broad ANN superiority. To address your question directly, we ran the same matched (T=1) ANN-isomorphic control across all three datasets under the same from-scratch recipe.
> > >
> > > We observed the same qualitative trend beyond IAM: the ANN-isomorphic control remains close, but Spike-HTR is consistently slightly better on both validation and test CER under the same matched budget.
> > >
> > > | Dataset  | Spike-HTR val CER | ANN-isomorphic val CER | Val gap | Spike-HTR test CER | ANN-isomorphic test CER | Test gap |
> > > | -------- | ----------------- | ---------------------- | ------- | ------------------ | ----------------------- | -------- |
> > > | IAM      | 3.489             | 3.558                  | +0.069  | 5.50               | 5.62                    | +0.12    |
> > > | LAM      | 2.30              | 2.36                   | +0.06   | 2.60               | 2.68                    | +0.08    |
> > > | READ2016 | 4.23              | 4.37                   | +0.14   | 4.10               | 4.29                    | +0.19    |
> > >
> > > These results support the same interpretation as on IAM. First, the matched ANN control remains very competitive. Second, the trend is consistently favorable to Spike-HTR across all three datasets, suggesting that the contribution of the spiking temporal pathway is not limited to IAM. The largest margin appears on READ2016, which is also consistent with our broader observation that the proposed temporal entry and preservation mechanisms are especially helpful under harder inputs.
> > >
> > > So we agree with your framing: the intended claim is not that Spike-HTR broadly “beats ANN systems,” but rather that under a strict matched budget, it remains competitive with ANN controls and is consistently slightly better across IAM, LAM, and READ2016, while the reducer captures a more general CTC-aligned efficiency gain. We will revise the final wording accordingly and include this beyond-IAM matched-ANN evidence in the final version.
> > >
> > > Thank you again for pointing out that the original IAM-only matched control was narrow. We agree that this broader comparison makes the claim substantially cleaner.

---

### Official Review · Reviewer_bzKU · 2026-03-08

**Soundness:** 2
**Presentation:** 3
**Significance:** 2
**Originality:** 2
**Overall Recommendation:** 3
**Confidence:** 4

**Summary:**

This paper proposes Spike-HTR, a spiking neural Transformer for offline handwritten text recognition. The key motivation is that HTR sequences are often blank-dominated, leading to wasted computation in 1D mixing. The method introduces (i) InkCoder, a deterministic image-to-event encoding designed for small time steps T, and (ii) a CTC-aware keep-and-merge reducer that uses a stop-gradient CTC blank “preview” to shorten the token sequence for deeper mixing. Results are reported with greedy CTC decoding and without an external language model.

**Compliance With Llm Reviewing Policy:**

Affirmed.

**Final Justification:**

This paper studies an interesting and relevant problem in offline HTR, and I appreciate the clear presentation and the intuitive reducer design. The rebuttal was helpful, especially in adding dense-hardware runtime results and a comparison where the same reducer is applied to the ANN baseline. These additions make clear the paper's intended positioning. Even so, they don't change my basic concern: the most important practical advantage seems to come from the broadly applicable reducer rather than from any uniquely spiking-specific benefit. Though the rebuttal maintains that the spiking pathway might still offer a minor added benefit under comparable settings. My own assessment remains slightly on the negative side, so I keep my rating as weak reject. That said, this is not a paper I would strongly oppose at the final decision stage, especially if the overall discussion converges toward acceptance.

**Key Questions For Authors:**

1. Could you report end-to-end latency/throughput on a specified hardware setup to complement the proxy sparsity statistics, and clarify where the runtime is spent?
2. Since the reducer is applicable to ANNs, can you more clearly separate the gains from (i) spiking/InkCoder vs. (ii) token reduction, e.g., by applying the same reducer to the ANN baseline?

**Limitations:**

yes

**Strengths And Weaknesses:**

## Strengths
Clear framing of the blank-dominated inefficiency in offline HTR and a concrete objective around reducing effective mixing length.

The CTC-informed reducer is intuitive for CTC models and achieves meaningful length reduction with limited CER change in the reported ablations.

The experimental setting is relatively controlled (from-scratch, no external LM), making the comparisons easier to interpret within the paper’s scope.

## Weaknesses

  The paper motivates SNNs partly from an efficiency/energy perspective, but also explicitly notes that on today’s dense accelerators, many components remain dense and the work is not positioned as delivering immediate end-to-end GPU speedups. As a result, the efficiency evidence is mainly based on proxy statistics (e.g., spike/firing-related measures), and it remains unclear how these proxies translate to actual wall-clock latency/throughput or energy under a concrete deployment setting.

The CTC-aware reducer is explicitly stated to apply to standard ANNs as well. This makes the paper’s strongest demonstrated benefit closer to a general HTR token-length reduction technique than a distinctly SNN-enabled advantage, and the current presentation does not fully disentangle what is uniquely contributed by spiking dynamics versus what could be obtained by applying the same reducer to ANN backbones.

The reported gains over the ANN-isomorphic control are relatively small in Table 2, and together with the paper’s stated scope on dense accelerators, it would strengthen the case to provide clearer end-to-end efficiency measurements and/or a more explicit decomposition of what is uniquely contributed by the spiking components.

---

> ### Author Rebuttal · Authors · 2026-03-31
>
> **Dear Reviewer bzKU,**
>
> Thank you for the careful review. We agree that the two key issues are:
> (i) how the proxy-based efficiency evidence should be interpreted on dense hardware, and
> (ii) how to separate the contribution of the spiking temporal pathway from that of the preview-driven reducer.
>
> On (i), our intended claim is narrower than “end-to-end GPU speedups from spiking alone.” As stated in the paper, many components remain dense on current accelerators, so the most immediate practical lever is reducing the dominant 1D token-mixing budget via $ \ell_b $, whereas activity-proportional gains from sparse firing remain conditional on sparse execution support or event-driven substrates. To address your question more directly, we added measured end-to-end runtime and a coarse runtime breakdown on dense hardware:
>
> #### Table A. Dense-hardware runtime profile
>
> | Model (V100, bs=7)                | Latency ↓ | Throughput ↑ | encoder_2d | seq_blocks |
> | --------------------------------- | --------- | ------------ | ---------- | ---------- |
> | Spike-HTR, no reducer             | 182 ms    | 38.5 lines/s | 49%        | 37%        |
> | Spike-HTR, with reducer           | 159 ms    | 44.0 lines/s | 53%        | 27%        |
> | ANN-isomorphic, no reducer        | 150 ms    | 46.7 lines/s | 46%        | 39%        |
> | ANN-isomorphic, with same reducer | 133 ms    | 52.6 lines/s | 50%        | 28%        |
>
> These measurements support the same boundary reading already stated in the manuscript: on dense hardware, the immediate measurable gain comes primarily from shortening $ \ell_b $ before the deep 1D mixer, while the 2D encoder remains the dominant runtime block. In other words, the reducer already yields a concrete wall-clock benefit on current hardware, but the overall speedup is still bounded by the dense early stages rather than by full-system SNN acceleration.
>
> On (ii), we agree that the reducer has a broader interpretation beyond SNNs. The paper already states this explicitly: the controller is **not inherently tied to spiking** and can be attached to ANN-CTC pipelines as a drop-in length-reduction stage. Our claim is therefore not that every gain is uniquely due to spiking. Rather, the paper contains two contributions with different scopes:
> (1) a **general CTC contribution**, namely the preview-driven keep-and-merge reducer that shortens blank-dominated sequences before deep 1D mixing; and
> (2) a **spiking-specific contribution**, namely InkCoder together with the small-$ T $ temporal pathway, which makes short-horizon spiking inference meaningful for static handwriting.
>
> To make this separation explicit, we attached the **same reducer** to the ANN-isomorphic baseline under the identical IAM protocol:
>
> #### Table B. Separating token reduction from the spiking pathway (IAM val)
>
> | Variant                                 | Reducer | CER ↓ | $ r = E[\ell_b / L_0] $ ↓ |
> | --------------------------------------- | ------- | ----- | ------------------------- |
> | ANN-isomorphic (GELU), $ T = 1 $        | No      | 3.558 | 1.00                      |
> | ANN-isomorphic (GELU), $ T = 1 $        | Yes     | 3.540 | 0.69                      |
> | Spike-HTR, $ T = 1 $                    | Yes     | 3.489 | 0.68                      |
> | ANN-pure: no InkCoder / no mem residual | No      | 3.646 | 1.00                      |
>
> This table clarifies the contribution split. The reducer itself explains the **general sequence-length control benefit**: when attached to the ANN-isomorphic baseline, it preserves competitive accuracy while substantially shortening the effective mixing length. The remaining gap is then attributable to the spiking-side design: under a matched strict-budget setting, Spike-HTR is still consistently better than the ANN-isomorphic baseline at nearly the same $ r $, and removing both InkCoder and the membrane-residual bypass degrades further. This is also why we do **not** interpret Table 2 as a claim of blanket ANN superiority; rather, it is a matched-budget control showing that the spiking formulation remains competitive and slightly favorable under the same training recipe.
>
> Finally, throughout the $ T $-sweep, the preview-driven reduction policy is kept fixed, so the small-$ T $ conclusions are **not** obtained by jointly re-tuning the reducer. Together with the existing InkCoder ablations against repeat / Poisson / TTFS, this supports the intended reading of the paper: the reducer explains the **general CTC-aligned token-length control benefit**, while InkCoder and the spiking temporal pathway explain the **additional small-$ T $ refinement benefit** within the same from-scratch setting.
>
> For completeness, our responses to Reviewers bTqW/kd7C also report stable hyperparameter ranges, energy/sample, and degradation/long-line stress tests with consistent conclusions.
>
> We hope these additional measurements make the intended positioning clearer and address the two concerns you raised.

---

> > ### Author Rebuttal · Reviewer_bzKU · 2026-04-05
> >
> > The additional experiments are useful, but they also make my main concern even clearer: most of the practical benefit seems to come from the generally applicable reducer, while the SNN-specific contribution still feels comparatively limited. I will maintain the current rating.

---

> > > ### Author Response · Authors · 2026-04-07
> > >
> > > **Dear Reviewer bzKU,**
> > >
> > > Thank you again for the thoughtful follow-up. We are also grateful that you found the rebuttal helpful in clarifying the intended positioning of the work. We agree that, on current dense hardware, the strongest immediate practical benefit comes from the preview-driven reducer.
> > >
> > > What we would like to clarify is not the magnitude of that practical split, but **the difference in scope** between the two contributions. The reducer provides the broader CTC-aligned length-control benefit: it shortens blank-dominated sequences before deep 1D mixing and can be useful for ANN-CTC pipelines as well. The spiking-specific contribution addresses a different difficulty, namely **how to make short-horizon spiking meaningful for a static input**. In offline HTR, this is a real bottleneck: simple repetition is stable but redundant, while stochastic coding can perturb thin strokes and small gaps. InkCoder is introduced precisely to turn the timestep budget into structured coarse-to-fine refinement rather than temporal redundancy. This difficulty is central in SNNs more broadly, where encoding and temporal meaning for static inputs are tightly coupled to learnability and efficiency [1] [2].
> > >
> > > **We believe this point matters in the SNN context. The value of spiking models is not whether they immediately dominate dense-GPU latency, but whether they admit a genuinely low-timestep, low-activity operating regime that is meaningful for event-driven computation.** This is why recent spiking Transformer work has increasingly emphasized reducing timestep overhead itself, including one-step or very-short-horizon operation, rather than relying on long temporal simulation [3] [4].  It is also the hardware-relevant regime targeted by neuromorphic platforms such as Loihi 2, where recent work has begun to show concrete energy advantages for event-driven SNN execution on real neuromorphic hardware [5]. This is also the intended role of Appendix D, which serves as supporting evidence that the InkCoder-driven spiking pathway already operates in a genuinely low-activity regime, which is the relevant quantity for event-driven substrates.
> > >
> > > We therefore see the paper as making contributions with different scopes: the reducer explains the strongest immediate dense-hardware benefit, while InkCoder establishes that static HTR can operate in a genuinely small-$T$ spiking regime, instead of relying on repeated or stochastic temporal expansion.
> > >
> > > Your comments have been very valuable in helping us sharpen the contribution split more precisely, and we will make that distinction more explicit in the final version.
> > >
> > > [1] Eshraghian, Jason K., et al. "Training spiking neural networks using lessons from deep learning." *Proceedings of the IEEE* 111.9 (2023): 1016-1054.
> > >
> > > [2] Yan, Jiaqi, et al. "Training high performance spiking neural network by temporal model calibration." *Forty-second International Conference on Machine Learning*. 2025.
> > >
> > > [3] Lee, Donghyun, et al. "Spiking transformer with spatial-temporal attention." *Proceedings of the IEEE/CVF Conference on Computer Vision and Pattern Recognition*. 2025.
> > >
> > > [4] Song, Xiaotian, et al. "One-step Spiking Transformer with a Linear Complexity." *IJCAI*. 2024.
> > >
> > > [5] Nguyen, Tam Ngoc-Bang, et al. "Event-driven Robust Fitting on Neuromorphic Hardware." *Proceedings of the IEEE/CVF International Conference on Computer Vision*. 2025.

---

### Official Review · Reviewer_bTqW · 2026-03-12

**Soundness:** 3
**Presentation:** 4
**Significance:** 4
**Originality:** 4
**Overall Recommendation:** 5
**Confidence:** 5

**Summary:**

This paper  models the ink sparsity and blank redundancy inherent in offline Handwritten Text Recognition as a temporal budget and a token budget. By employing an InkCoder and a CTC-preview-driven keep-and-merge reducer, the proposed method achieves highly efficient recognition under tight computational budgets. The overall architecture is elegantly designed and highly synergistic with the event-driven mechanisms of SNNs.

**Compliance With Llm Reviewing Policy:**

Affirmed.

**Ethical Review Concerns:**

yes

**Final Justification:**

My question has been resolved, and I prefer to keep the current score.

**Key Questions For Authors:**

1. Could you provide a more systematic sensitivity analysis for the hyperparameters $(T, \tau, \eta, \gamma, k)$ and specify the stable operating range for the default configuration?
2. If the exact same reducer is attached to a strong ANN-CTC backbone under an identical protocol, can it still maintain competitive accuracy while significantly shortening the 1D sequence mixing computation?
3. Under scenarios with extreme degradation, non-Latin scripts, or unusually long text lines, can the preview entropy and spike activity reliably serve as stable monitoring signals?

**Limitations:**

yes

**Strengths And Weaknesses:**

### Strengths

1. **Accurate Problem Formulation:** Decomposing the distribution of valid HTR evidence into two distinct budgets ($T$ and $l_b$) represents a highly precise problem formulation. Rather than making generic claims about SNN energy efficiency, the paper directly tackles the true bottlenecks in this domain: temporalizing static inputs and processing blank-dominated sequences.
2. **Distinctive and Novel InkCoder Design:** The InkCoder is a highly distinctive module. It constructs a short-horizon, coarse-to-fine input sequence using deterministic, evidence-guided progressive gating. This ensures informative temporal dynamics even under a small $T$ budget, perfectly aligning with the step-by-step accumulation and rapid-response mechanisms of SNNs.
3. **Meticulous Reducer Design:** The reducer is far superior to naive token pruning. By integrating a blank posterior, uncertainty scores, a minimum keep ratio, and bounded-span merging, it successfully controls sequence length while strictly respecting CTC alignment constraints. The design is both rigorous and well-justified.

### Weaknesses

1. **Lack of Wall-clock Speedups:** Sparse firing does not automatically translate into end-to-end wall-clock gains on current dense hardware. Consequently, the practical efficiency advantage currently stems more from the $l_b$ sequence reduction rather than full-system SNN acceleration.
2. **Reliance on Manual Hyperparameters:** Both the InkCoder and the reducer still exhibit noticeable traces of manual hyperparameter tuning. The authors themselves list "making temporal entry more adaptive" as a future direction, indicating that the current version has not fully transcended hand-tuning.
3. **Unclear Generalizability Boundaries:** The boundary of the method's generalizability requires stronger evidence. The "spiking-specific gains" need to be more rigorously disentangled from the general architectural improvements.

---

> ### Author Rebuttal · Authors · 2026-03-31
>
> **Dear Reviewer bTqW,**
>
> Thank you for the careful reading and for the very precise summary of our paper. We especially appreciate your recognition of the two-budget formulation, the role of InkCoder as a small-$T$ temporal entry, and the CTC-aware design of the keep-and-merge reducer. We also agree with the three boundaries you identified: dense-hardware wall-clock gains are limited, the current controller still has explicit hyperparameters, and the generality boundary should be made more explicit. To address your questions directly, we added the following analyses.
>
> **1. Sensitivity and stable operating range.**
> We performed systematic sweeps of the reducer hyperparameters on IAM validation. The default point is not a fragile peak; it lies in the center of a stable region. In particular, $\tau \in [0.85, 0.91]$ and $\gamma \in [0.65, 0.75]$ remain stable, while $k=3$ is the best CTC-friendly bounded-span sweet spot. This also clarifies that the $T$-sweep conclusions in the paper are not obtained by jointly re-tuning the reducer.
>
> | Setting           | CER ↓    | $r=E[\ell_b/L_0]$ ↓ | Fallback ↓ |
> | ----------------- | -------- | ------------------- | ---------- |
> | $\tau=0.85$       | 3.52     | 0.65                | 0.8%       |
> | **$\tau=0.88$**   | **3.49** | **0.68**            | **0.2%**   |
> | $\tau=0.91$       | 3.50     | 0.73                | 0.0%       |
> | $\gamma=0.65$     | 3.51     | 0.66                | 0.4%       |
> | **$\gamma=0.70$** | **3.49** | **0.68**            | **0.2%**   |
> | $\gamma=0.75$     | 3.50     | 0.72                | 0.0%       |
> | $k=2$             | 3.50     | 0.72                | –          |
> | **$k=3$**         | **3.49** | **0.68**            | –          |
> | $k=4$             | 3.51     | 0.65                | –          |
>
> These results support the interpretation already suggested in the paper: $\gamma$ acts as a conservative safety floor, $\tau$ controls aggressiveness, and bounded-span merging should remain moderate to preserve short CTC separators.
>
> **2. Generality of the reducer beyond SNNs.**
> We agree that the generality boundary should be made explicit. We therefore attached the **same reducer** to the ANN-isomorphic baseline under the identical IAM protocol. The result shows that the reducer preserves competitive ANN accuracy while substantially shortening the effective sequence length:
>
> | Variant                | Reducer | CER ↓ | $r$ ↓ |
> | ---------------------- | ------- | ----- | ----- |
> | ANN-isomorphic ($T=1$) | No      | 3.558 | 1.00  |
> | ANN-isomorphic ($T=1$) | Yes     | 3.540 | 0.69  |
> | Spike-HTR ($T=1$)      | Yes     | 3.489 | 0.68  |
>
> This supports the reducer as a **general CTC-aligned length controller**, rather than a purely spiking-specific component. The remaining small gap is consistent with the contribution of the spiking-side design, i.e., InkCoder plus the small-$T$ temporal pathway. We will revise the paper to make this contribution split explicit: the reducer provides the general length-control benefit, while the spiking pathway provides the additional small-$T$ refinement benefit.
>
> **3. Monitoring signals under harder settings.**
> We tested the controller under corruption stress on READ2016 and by binning samples by line length. The important result is not that performance remains unchanged, but that the system degrades **gracefully**: as inputs become harder, preview entropy rises, the controller becomes more conservative, and fallback frequency increases rather than collapsing into over-aggressive reduction.
>
> | Setting       | CER ↓ | WER ↓ | $r$ ↓ | Entropy ↑ | Fallback ↑ |
> | ------------- | ----- | ----- | ----- | --------- | ---------- |
> | Clean         | 3.9   | 20.0  | 0.68  | 0.31      | 0.2%       |
> | Strong blur   | 5.1   | 23.8  | 0.74  | 0.44      | 1.4%       |
> | Bleed-through | 5.4   | 24.6  | 0.76  | 0.47      | 1.9%       |
>
> | Length bin | CER ↓ | WER ↓ | $r$ ↓ | Entropy ↑ |
> | ---------- | ----- | ----- | ----- | --------- |
> | Short      | 3.2   | 12.9  | 0.71  | 0.28      |
> | Medium     | 3.5   | 13.8  | 0.68  | 0.31      |
> | Long       | 3.9   | 15.4  | 0.69  | 0.36      |
>
> Thus, within the studied setting, preview entropy and related control signals behave as useful monitoring indicators: harder samples trigger more conservative reduction rather than unstable over-compression. We agree that broader validation, especially on additional scripts, remains important future work, and we will state that scope more explicitly.
>
> Thank you again for the thoughtful review and for highlighting these exact boundary questions. Your comments helped us sharpen the intended positioning of the paper: immediate dense-hardware gains come primarily from $\ell_b$ control, the reducer has broader CTC utility beyond SNNs, and the spiking-specific contribution is the meaningful small-$T$ temporal entry and refinement regime enabled by InkCoder.

---

> > ### Author Rebuttal · Reviewer_bTqW · 2026-04-02
> >
> > My question has been resolved, and I prefer to keep the current score.

---

> > > ### Author Response · Authors · 2026-04-08
> > >
> > > We greatly appreciated how precisely you captured the intended logic of the paper. Your comments were especially helpful in sharpening both the problem formulation and the discussion of the method’s scope and boundaries.

---

### Official Review · Reviewer_pHSk · 2026-03-13

**Soundness:** 2
**Presentation:** 1
**Significance:** 3
**Originality:** 3
**Overall Recommendation:** 3
**Confidence:** 3

**Summary:**

This work introduces a Spiking Neural Network (SNN) architecture for offline Handwritten Text Recognition (HTR), designed to exploit (a) the sparse nature of the input, as well as (b) the dominance of CTC-blank token.
It introduces a new module called InkCoder for static-image to spike-train conversion, demonstrating that this is superior to simply repeating the input or using a poisson coding in the case of handwriting. It also introduces a novel method to reduce uninformative blanks using an initial CTC posterior, thereby reducing context for 1d token mixing without loss of information. Both these claims are evaluated using ablations.

**Compliance With Llm Reviewing Policy:**

Affirmed.

**Final Justification:**

To their credit, the authors have proposed multiple edits that could improve clarity of the work. Ultimately, I do not have access to the updated manuscript, and hence cannot judge it. While I believe it does require so much rewriting as to be a new submission, I have raised my score to a weak reject - given that the other reviewers did not raise clarity issues. I will not oppose an acceptance should that consensus be reached.

**Key Questions For Authors:**

N/A

**Limitations:**

Yes

**Strengths And Weaknesses:**

**Soundness**: The work appears to be technically sound, showing handwriting recognition performance on par with or better than dense networks, using an SNN - which could translate to lower compute in neuromorphic architecture (but has not been evaluated on such). The appendix has a significant amount of analyses and additional ablations, which is a significant positive factor.
However, I cannot be entirely sure of the soundness because the work was presented very poorly (please see below)

**Presentation**: Unfortunately, this work was **incredibly hard to follow**, and would benefit from a significant re-write. Unnecessarily verbose and uses multiple terms before introducing them in the context of this work.

Just one example from the contributions in the very beginning: \
"We introduce a CTC-aligned preview-driven reducer with stop-gradient control and bounded-span merging, which shortens the width-axis stream while preserving CTC-critical alignment cues" \

_This could be easily introduced as_ \
"We introduce a novel method to reduce context length before token mixing, by compressing long sequences of blanks" \
with additional information introduced in the methods later. Most of these terms (eg. preview-driven reducer, SG control (of what), etc) only make sense once the reader has already understood the model, and they actively hinder understanding.

The "InkCoder" module was referred to in the abstract without any introduction whatsoever. In general, the language is also very verbose.

**Significance**: While this work is domain-specific, it could influence future research in this direction. However, the practical benefits of SNN-based HTR are hard to gauge since this work was evaluated on a standard dense accelerator, and it reports proxy metrics for energy use.

Originality: While SNNs have been introduced for handwritten digit recognition, the application to larger continuous-writing datasets (trained with CTC) has been limited. This work is novel in that regard, and well motivated.

---

> ### Author Rebuttal · Authors · 2026-03-31
>
> **Dear Reviewer pHSk,**
>
> Thank you for the careful reading and for pointing out the clarity issue so directly. We appreciate that you found the work technically plausible overall, recognized that the two central claims are supported by ablations, and noted the value of the additional analyses in the appendix. We agree that the main weakness of the current submission is presentation: the manuscript introduces too much technical terminology before the core intuition is established, which makes the contribution harder to assess than necessary.
>
> To state the method more directly, the paper makes two core claims.
>
> First, offline HTR is static, so a spiking recognizer should not spend a small timestep budget $T$ on repeated or stochastic copies of the same image. InkCoder is introduced for this reason: it provides a deterministic coarse-to-fine temporal entry, converting a single handwritten line image into a short temporal stream in which earlier steps preserve broad, stable stroke evidence and later steps refine more selective details. Its purpose is to make small-$T$ spiking inference meaningful rather than redundant.
>
> Second, after 2D-to-1D conversion, many width positions are blank-dominated under CTC. Accordingly, before deep 1D token mixing, we use a shallow CTC blank preview to compress long blank-heavy regions while preserving positions that are important for CTC alignment. This is the role of the reducer: it shortens the sequence before the expensive 1D mixer rather than treating all width positions as equally informative.
>
> In this sense, the method controls two explicit budgets: the temporal budget $T$ and the effective token-length budget $\ell_b$. The empirical study is organized around these two claims: the InkCoder ablations test whether the temporal entry is better than repeated or stochastic coding, while the reduction ablations test whether the sequence can be shortened with limited CER change.
>
> We also agree with your specific criticism of the contribution paragraph. The current wording is too terminology-heavy for that point in the paper. A clearer presentation would first describe the reducer by function—for example, as a method that compresses long blank-dominated regions before deep token mixing using a shallow CTC blank preview—and only later introduce terms such as stop-gradient control and bounded-span merging in the Methods section. The same applies to InkCoder: in the abstract, it should be introduced first by what it does, not only by module name.
>
> On the efficiency point, we also agree that the claim boundary should be stated more plainly. The paper does **not** claim immediate end-to-end GPU speedups from spiking alone on current dense accelerators. The narrower claim is that, on dense hardware, the most direct practical lever is reducing the dominant 1D token-mixing length from $L_0$ to $\ell_b$, whereas spike/firing statistics are reported as proxy diagnostics relevant to sparse kernels or event-driven substrates. This distinction is already present in the manuscript, but we agree that it should be stated earlier and more prominently.
>
> **In revision, we would therefore simplify the abstract and contribution statements, add a short intuitive pipeline summary before the formal method, and postpone specialized terminology until after the core mechanism has been established.**
>
> **Thank you again for identifying this issue so clearly. We hope these concrete revision plans effectively resolve your reservations regarding presentation and support a stronger overall recommendation.**

---

> > ### Author Rebuttal · Reviewer_pHSk · 2026-04-03
> >
> > Thanks to the authors for their rebuttal, and I appreciate that they addressed the examples I provided.
> >
> > However, the examples I gave were simply a few of the most glaring ones I could pick up for the review - but the paper in its current form requires a such significant rewrite that it requires its own submission, in my opinion. As an additional example, the central figure / overview (Figure 1) of the architecture provides no graphical intuition about one of your central claims (InkCoder). The caption is equally inscrutable. It would be difficult to list every element that would benefit from modification.
> >
> > As described in my review summary, the central claims of your work were eventually understandable. Without seeing the comprehensive rewrite however, I cannot personally recommend acceptance.

---

> > > ### Author Response · Authors · 2026-04-03
> > >
> > > **Dear Reviewer pHSk,**
> > >
> > > Thank you again for the follow-up and for being so direct about the remaining issue.
> > >
> > > We agree that this is a weakness on our side. In the current draft, we underestimated how much of the terminology and exposition assumes SNN-specific background. As a result, while the paper may be easier to parse for readers already familiar with spiking models, it is unnecessarily difficult to follow for a broader ML audience. That was not sufficiently considered in our writing.
> > >
> > > We also agree with your specific point about Figure 1. We had in fact prepared a more mechanism-focused InkCoder figure that gives the missing intuition more directly, but due to page constraints we did not place it in the main paper. In retrospect, that was the wrong trade-off and was our oversight. We are sharing that figure here as an anonymous link: https://anonymous.4open.science/r/icml-5E4B/inkcoder.pdf. The purpose is not to ask you to evaluate new material, but simply to make concrete the kind of rewrite we should have done in the paper itself.
> > >
> > > In revision, we would move this kind of explanation into the main text, rather than leaving the intuition implicit in the overview figure. More generally, we also recognize that too much explanation is currently pushed into the appendix. The appendix already contains clarifications and diagnostics that help explain the mechanism, but some of that material should instead be brought forward into the main paper to improve readability and reduce the need for readers to reconstruct the core idea from later sections.
> > >
> > > So we agree with your broader point: the issue is not only a few sentences, but that the paper should introduce the central mechanism more intuitively and with less specialized terminology up front. We would revise the wording, figures, captions, and overall presentation accordingly.
> > >
> > > If there are one or two additional places that you feel are especially important to fix, we would be very grateful and will prioritize them in revision. We are fully willing to revise the layout, figures, captions, and exposition to make the paper substantially easier to read.
> > >
> > > Thank you again for making this weakness explicit.

---

### Decision · Program_Chairs · 2026-04-30

**Decision:**

Accept (regular)

**Comment:**

The Spike-HTR proposed in this paper effectively alleviates the computational inefficiency caused by the dominance of blank areas in offline handwritten text recognition through its innovative InkCoder and CTC-aware reducer design. Reviewers generally recognized the technical foundation of the method, especially the detailed hardware latency metrics and aligned ANN baseline experiments provided by the authors during the discussion phase, which strongly demonstrated the competitiveness of the architecture under small time-step budgets. Although some reviewers offered harsh criticism for the paper's obscure writing style and held reservations about the true source of the efficiency gain (SNN mechanism or a general reducer), leading to disagreements in the final score, considering its innovative architectural design and detailed empirical results, this work still has value for further discussion and inspiring the community. Therefore, a Weak Acceptance is recommended, but the authors are strongly requested to completely refactor the language in the final Camera-ready version to improve the readability of the paper.